# Analysis of a Fractional-Order COVID-19 Epidemic Model with Lockdown

**DOI:** 10.3390/vaccines10111773

**Published:** 2022-10-22

**Authors:** Dawit Denu, Seth Kermausuor

**Affiliations:** 1Department of Mathematical Sciences, Georgia Southern University, Savannah, GA 31419, USA; 2Department of Mathematics and Computer Science, Alabama State University, Montgomery, AL 36101, USA

**Keywords:** COVID-19 epidemic model, lockdown, fractional-order differential equations, stability of equilibrium solutions, fractional power series

## Abstract

The outbreak of the coronavirus disease (COVID-19) has caused a lot of disruptions around the world. In an attempt to control the spread of the disease among the population, several measures such as lockdown, and mask mandates, amongst others, were implemented by many governments in their countries. To understand the effectiveness of these measures in controlling the disease, several mathematical models have been proposed in the literature. In this paper, we study a mathematical model of the coronavirus disease with lockdown by employing the Caputo fractional-order derivative. We establish the existence and uniqueness of the solution to the model. We also study the local and global stability of the disease-free equilibrium and endemic equilibrium solutions. By using the residual power series method, we obtain a fractional power series approximation of the analytic solution. Finally, to show the accuracy of the theoretical results, we provide some numerical and graphical results.

## 1. Introduction and Preliminaries

The study of mathematical models of infectious diseases has attracted the attention of many researchers since they provide a better understanding of their evaluations, existence, stability, and control [1,2,3,4,5,6]. Most of the mathematical models of infectious diseases are composed of a system of integer-order differential equations. However, in the last few decades, fractional-order differential equation has been used in the modeling of biological phenomena because they provide a greater degree of accuracy than the integer-order models [7,8,9,10,11,12,13]. The theory of fractional calculus, which involves differentiation and integration of non-integer orders, is as old as the classical calculus of integer orders. However, this theory has gained considerable attention from many researchers in recent years due to the numerous applications found in the sciences, engineering, economics, control theory, and finance, amongst others. The increasing interest in using fractional calculus in the modeling of real-world phenomena is due to its various properties which are not found in classical calculus. Unlike the integer-order derivatives, which are local in nature, most of the fractional-order derivatives (for example, the Caputo and Riemman–Liouville fractional order derivatives [14,15]) are non-local and possess the memory effects which make it more superior because in many situations the future state of the model depends not only upon the current state but also on the previous history. For this realistic property, the usage of fractional-order systems is becoming popular to model the behavior of real systems in various fields of science and engineering. Many researchers have therefore expanded the integer-order models to fractional-order models via various mathematical techniques. Recently, several authors have used fractional-order models to understand the dynamics of the COVID-19 epidemic to achieve the best strategies to stop the spread of the disease, chose a better effective immunization program, allocate scarce resources to control or prevent infections and also predict the future course of the outbreak [16,17,18,19,20,21,22,23,24,25,26,27].

Motivated by the above works and the high interest in finding the best solution to control the COVID-19 pandemic, we study a fractional-order model to understand the dynamics of the COVID-19 infections when the population is under lockdown. Most of the COVID-19 models in the literature, to the best of our knowledge, do not include the impact of the preventive measures adopted by many governments around the world to control the virus. Lockdown was one of the preventive measures and that makes our model quite unique and interesting.

Several definitions of fractional derivatives and integrals have been provided in the literature. For the purpose of this work, we present the definitions and some properties of the Riemann–Liouville fractional integral, Caputo fractional derivative and other related fractional-order derivatives. For more information about the Riemann–Liouville fractional integral and the Caputo derivative, we refer the interested reader to [14,15] and the references therein.

**Definition** **1**([14]). *The Riemann–Liouville fractional integral of order α>0 of a function f is given by*
(1)$$\begin{array}{c} J_{a}^{\alpha }f\left(t\right)=\frac{1}{\Gamma (\alpha )}\int_{a}^{t}\frac{f(z)}{{\left(t-z\right)}^{1-\alpha }}dz, \end{array}$$*where* Γ* denotes the Gamma function defined by*
$$\begin{array}{c} \Gamma \left(x\right)=\int_{0}^{\infty }t^{x-1}e^{-t}dt,\;\;x>0. \end{array}$$

**Definition** **2**([14]). *The Caputo fractional derivative of order α>0 is given by*
(2)$$\begin{array}{c} D_{a}^{\alpha }f\left(t\right)=\frac{1}{\Gamma (n-\alpha )}\int_{a}^{t}\frac{f^{\left(n\right)}\left(z\right)}{{\left(t-z\right)}^{\alpha -n+1}}dz,\;t>a,n-1<\alpha \leq n,\;n\in N. \end{array}$$

The Caputo derivative satisfies the following properties:1.$D_{a}^{\alpha }\left(b_{1}f\left(t\right)+b_{2}g\left(t\right)\right)=b_{1}D_{a}^{\alpha }f\left(t\right)+b_{2}D_{a}^{\alpha }g\left(t\right),\;\;b_{1},b_{2}\in R$.2.$D_{a}^{\alpha }\left(J_{a}^{\alpha }f\left(t\right)\right)=f\left(t\right).$3.$J_{a}^{\alpha }\left(D_{a}^{\alpha }f\left(t\right)\right)=f\left(t\right)-\sum_{k=0}^{n-1}\frac{f^{\left(k\right)}\left(a\right)}{k!}{\left(t-a\right)}^{k}$.4.$D_{a}^{\alpha }\left(c\right)=0,\;\;c\in R$.5.$D_{a}^{\alpha }{\left(t-a\right)}^{\lambda }=\frac{\Gamma (\lambda +1)}{\Gamma (\lambda -\alpha +1)}{\left(t-a\right)}^{\lambda -\alpha }$, for λ>n−1.6.$D_{a}^{\alpha }{\left(t-a\right)}^{k}=0$, for k=0,1,2,⋯,n−1.

Most fractional-order differential equations do not have a closed-form solution, and thus approximation and numerical methods are extensively used. One way to find an approximated solution of a system of fractional-order differential equations is by using the technique of fractional power series. The following is the definition of the fractional power series.

**Definition** **3**([28]). *The fractional power series about $t=t_{0}$ is defined as*
(3)$$\begin{array}{c} \sum_{m=0}^{\infty }c_{m}{\left(t-t_{0}\right)}^{m\alpha }=c_{0}+c_{1}{\left(t-t_{0}\right)}^{\alpha }+c_{2}{\left(t-t_{0}\right)}^{2\alpha }+\cdots , \end{array}$$*where n−1<α≤n for some n∈N and $c_{m}$ for m=0,1,2,⋯ are the coefficients of the power series.*

The following result hold for the Caputo derivative.

**Theorem** **1**([28]). *Suppose the fractional power series representation of f is of the form*
$$\begin{array}{c} f\left(t\right)=\sum_{m=0}^{\infty }c_{m}{\left(t-t_{0}\right)}^{m\alpha }. \end{array}$$
*If $D_{a}^{m\alpha }f\left(t\right)$ for m=0,1,2,3⋯ are continuous on the interval $\left(t_{0},t_{0}+\rho \right)$, then $c_{m}=\frac{D_{a}^{m\alpha }f\left(t_{0}\right)}{\Gamma (1+m\alpha )}$, where $D_{a}^{m\alpha }=D_{a}^{\alpha }D_{a}^{\alpha }\cdots D_{a}^{\alpha }$ (m-times) and ρ is the radius of convergence.*


**Remark** **1.**
*We note that the kernel function in the definition of the Caputo derivative ${\left(t-z\right)}^{n-1-\alpha }$ has a singularity at z=t. Recently, some new fractional derivatives with non singular kernels were introduced in the literature and we present them here for the readers’ reference.*


In 2015, Caputo annd Frabrizio [29] proposed the following generalization of the Caputo derivative as follows:

**Definition** **4.**
*The Caputo–Fabrizio fractional-order derivative (CF) of order α is defined as follows:*

(4)
$$\begin{array}{c} {}^{CF}D_{a}^{\alpha }f\left(t\right)=\frac{M(\alpha )}{\left(1-\alpha \right)}\int_{a}^{t}f^{\prime }\left(z\right)\exp\left\{-\frac{\alpha (t-z)}{1-\alpha }\right\}dz,\;\;\;\;\;\;0<\alpha \leq 1 \end{array}$$

*where M(α) is a normalization function such that M(0)=M(1)=1. The Caputo–Fractional derivative of a constant function is zero, but unlike the Caputo derivative in (Equation 2), the kernel does not have a singularity at z=t.*


The Caputo–Fabrizio fractional-order had also been generalized by Atangana and Baleanu in 2016 as follows:

**Definition** **5**([30]). *Let $f\in H^{1}\left(a,b\right),\;a<b$ and α∈[0,1]. The Atangana–Baleanu fractional-order derivative in the Caputo sense is given by*
(5)$$\begin{array}{c} {}^{ABC}D_{a}^{\alpha }f\left(t\right)=\frac{B(\alpha )}{\left(1-\alpha \right)}\int_{a}^{t}f^{\prime }\left(z\right)E_{\alpha }\left\{-\frac{\alpha (t-z)}{1-\alpha }\right\}dz \end{array}$$*where B(α) is a normalization function such that B(0)=B(1)=1, $E_{\alpha }\left(\cdot \right)$ is the one parameter Mittag–Leffler function (see Remark 3) and $H^{1}\left(a,b\right)$ is the Sobolev space of order one defined as follows:*
$$\begin{array}{c} H^{1}\left(a,b\right)=\left\{u\in L^{2}\left(a,b\right):u^{\prime }\in L^{2}\left(a,b\right)\right\}. \end{array}$$

We refer the interested reader to [29,30] and the references therein for more information about these fractional derivatives.

**Remark** **2.**
*It is worth pointing out that recently, Hattaf [31] also introduced some new generalizations of the Attangana–Baleanu fractional-order derivatives by introducing a weight function and thus giving rise to the definition of several fractional-order derivatives with non-singular kernels.*


Next, we present the definition of the two-parameter Mittag-Leffler function, which will be utilized later in the paper.

**Definition** **6**([28,32]). *The two-parameter Mittag-Leffler function denoted by $E_{\alpha ,\beta }$, is defined as*
(6)$$E_{\alpha ,\beta }\left(z\right)=\sum_{k=0}^{\infty }\frac{z^{k}}{\Gamma (\alpha k+\beta )},\;Re\left(\alpha \right),Re\left(\beta \right)>0,\beta \in C.$$

**Remark** **3.**
*When β=1 in (Equation 6), then we have the one parameter Mittag-Leffler function which is denoted by $E_{\alpha }\left(\cdot \right)$. That is, $E_{\alpha }\left(z\right)=E_{\alpha ,1}\left(z\right)$*


## 2. Model Formulation

In order to understand the impact of lockdown in preventing the spread of COVID-19 infection, Baba et al. [33] proposed the following model governed by a system of nonlinear ordinary differential equations:(7)$$\begin{array}{c} \frac{dS}{dt}=\Lambda -\beta SI-\lambda_{1}SL-dS+\gamma_{1}I+\gamma_{2}I_{L}+\theta_{1}S_{L},\\ \frac{dS_{L}}{dt}=\lambda_{1}SL-dS_{L}-\theta_{1}S_{L},\\ \frac{dI}{dt}=\beta SI-\gamma_{1}I-\alpha_{1}I-dI-\lambda_{2}IL+\theta_{2}I_{L},\\ \frac{dI_{L}}{dt}=\lambda_{2}IL-dI_{L}-\theta_{2}I_{L}-\gamma_{2}I_{L}-\alpha_{2}I_{L},\\ \frac{dL}{dt}=\mu I-\phi L, \end{array}$$
subjected to the initial conditions
$$S\left(0\right)=S_{0}>0,\;\;S_{L}\left(0\right)=S_{L}^{0}>0,\;\;I\left(0\right)=I^{0}>0,\;\;I_{L}\left(0\right)=I_{L}^{0}>0\;\;\mathrm{and}\;\;L\left(0\right)=L_{0}>0.$$

In this model, the authors considered a population of total size N(t) at time *t* with a constant recruitment rate of Λ. The population is divided into four compartments denoted by $S\left(t\right),S_{L}\left(t\right),I\left(t\right)$ and $I_{L}\left(t\right)$. The S(t) class denotes the susceptible individuals that are not under lockdown. The group $S_{L}\left(t\right)$ contains those individuals who are susceptible and are under lockdown. The group who are infected and are not under lockdown are represented by I(t), while those individuals who are infected and are under lockdown are denoted by $I_{L}\left(t\right).$ Finally, the cumulative density of the lockdown program is denoted by L(t). The authors have studied the local stability of the equilibrium solutions in relation to the basic reproduction number.

In order to include the memory effects and the past history to get a better understanding of the dynamics of COVID-19 infections under lockdown, we reformulate the model (Equation 7) by using the Caputo fractional derivative as follows:(8)$$\begin{array}{c} D_{0}^{\alpha }S\left(t\right)=\Lambda -\beta SI-\lambda_{1}SL-dS+\gamma_{1}I+\gamma_{2}I_{L}+\theta_{1}S_{L}\\ D_{0}^{\alpha }S_{L}\left(t\right)=\lambda_{1}SL-dS_{L}-\theta_{1}S_{L}\\ D_{0}^{\alpha }I\left(t\right)=\beta SI-\gamma_{1}I-\alpha_{1}I-dI-\lambda_{2}IL+\theta_{2}I_{L}\\ D_{0}^{\alpha }I_{L}\left(t\right)=\lambda_{2}IL-dI_{L}-\theta_{2}I_{L}-\gamma_{2}I_{L}-\alpha_{2}I_{L}\\ D_{0}^{\alpha }L\left(t\right)=\mu I-\phi L, \end{array}$$
where $D_{0}^{\alpha }$ denotes the Caputo derivative for 0<α≤1; under the initial conditions
$$S\left(0\right)=S_{0}>0,\;\;S_{L}\left(0\right)=S_{L}^{0}>0,\;\;I\left(0\right)=I^{0}>0,\;\;I_{L}\left(0\right)=I_{L}^{0}>0\;\mathrm{and}\;L\left(0\right)=L_{0}>0.$$

We explore some interesting results of the fractional-order COVID-19 model (Equation 8), description of the parameters shown in Table 1. In particular, after proving the existence of a unique positive global solution of the model, we study the local and global stability of the various equilibrium points of the fractional-order model using the comparison theory of fractional differential inequality and fractional La-Salle invariance principle. Moreover, we apply the method of residual power series technique to approximate the solution of the fractional-order system (Equation 8). Finally, we provide numerical simulations to illustrate some of the theoretical results and error analysis to show how good the approximated solution is.

**Remark** **4.**
*In [17], the authors also discussed a slightly different extension of the model in (Equation 7) to fractional calculus where they include the order of the fractional derivative as powers on the parameters. They established the existence and uniqueness of solutions to their model using the Schauder and Banach fixed point theorems.*


Denote the set Ω as follows,
$$\Omega =\left\{\left(S,S_{L},I,I_{L},L\right)\in R_{+}^{5}:S+S_{L}+I+I_{L}\leq \frac{\Lambda }{d}\;\mathrm{and}\;L\leq \frac{\mu \Lambda }{\phi d}\right\}.$$

**Theorem** **2.**
*For any t≥0 and X(0)∈Ω there exist a unique positive solution $X\left(t\right)=(S\left(t\right),S_{L}\left(t\right),I\left(t\right),I_{L}\left(t\right),L\left(t\right))$ of system (Equation 8). Moreover, the set Ω attracts all solutions of system (Equation 8), and thus it is positively invariant.*


**Proof.** Denote the vector field associated with system (Equation 8) by f(X(t)). Then f(X(t)) can be written as f(X(t))=A+(S(t)B+C+I(t)D)X(t), where
$$A=\left(\begin{array}{c} \Lambda \\ 0\\ 0\\ 0\\ 0 \end{array}\right)\;B=\left(\begin{array}{ccccc} 0 & 0 & -\beta & 0 & -\lambda_{1}\\ 0 & 0 & 0 & 0 & \lambda_{1}\\ 0 & 0 & \beta & 0 & 0\\ 0 & 0 & 0 & 0 & 0\\ 0 & 0 & 0 & 0 & 0 \end{array}\right)\;C=\left(\begin{array}{ccccc} 0 & 0 & 0 & 0 & 0\\ 0 & 0 & 0 & 0 & 0\\ 0 & 0 & 0 & 0 & -\lambda_{2}\\ 0 & 0 & 0 & 0 & \lambda_{2}\\ 0 & 0 & 0 & 0 & 0 \end{array}\right)$$
and
$$D=\left(\begin{array}{ccccc} -d & \theta_{1} & \gamma_{1} & \gamma_{2} & 0\\ 0 & -d-\theta_{1} & 0 & 0 & 0\\ 0 & 0 & -\gamma -1-\alpha_{1}-d & \theta_{2} & 0\\ 0 & 0 & 0 & -d-\theta -2-\gamma_{2}-\alpha_{2} & 0\\ 0 & 0 & \mu & 0 & -\phi \end{array}\right).$$Clearly, f(X(t)) and $\frac{\partial f(X)}{\partial X}$ are continuous for all X(t)∈Ω. Moreover,
$$\left||f\left(X\left(t\right)\right)|\right|\leq \left||A|\right|+\left||\left(S\left(t\right)B+I\left(t\right)C+D\right)X|\right|\leq k_{1}+k_{2}||X||,$$
where $k_{1}=\left||A|\right|$ and $k_{2}=\max\left\{1,\frac{\Lambda }{d}\right\}||B+C+D||$ are two positive constants. Thus by Theorem (3.1) and remark (3.2) of [34], the system (Equation 8) has a unique solution.Now, to prove that Ω is positively invariant, let $N\left(t\right)=S\left(t\right)+S_{L}\left(t\right)+I\left(t\right)+I_{L}\left(t\right)$. Then adding the first four equations of system (Equation 8), we have
$$D_{0}^{\alpha }N\left(t\right)=\Lambda -dN\left(t\right)-\alpha_{1}I-\alpha_{2}I_{L}.$$Taking the Laplace transform on both sides yields
$$L\left(D_{0}^{\alpha }N\left(t\right)\right)=\frac{\Lambda }{s}-dL\left(N\left(t\right)\right)-\alpha_{1}L\left(I\left(t\right)\right)-\alpha_{2}L\left(I_{L}\left(t\right)\right).$$Simplifying this equation, we have the following inequality
$$L\left(N\left(t\right)\right)\leq \frac{\Lambda s^{-1}}{s^{\alpha }+d}+\frac{s^{\alpha -1}N\left(0\right)}{s^{\alpha }+d}.$$Taking the inverse Laplace transform and using the fact that
$$L^{-1}\left[\frac{s^{-(\alpha -\beta )}}{s^{\beta }-a}\right]=t^{\alpha -1}E_{\beta ,\alpha }\left(at^{\beta }\right),\;\alpha ,\beta >0,\;s^{\alpha }>\left|a\right|,$$
where $E_{\alpha ,\beta }\left(.\right)$ is the Mittag-Leffler function defined in (Equation 6), we have
(9)$$\begin{array}{cc} N(t) & \leq \Lambda t^{\alpha }E_{\alpha ,\alpha +1}\left(-dt^{\alpha }\right)+N\left(0\right)E_{\alpha ,1}\left(-dt^{\alpha }\right)\\ \; & =\Lambda t^{\alpha }E_{\alpha ,\alpha +1}\left(-dt^{\alpha }\right)+N\left(0\right)\left(-dt^{\alpha }E_{\alpha ,\alpha +1}\left(-dt^{\alpha }\right)+\frac{1}{\Gamma (1)}\right)\\ \; & \leq \Lambda t^{\alpha }E_{\alpha ,\alpha +1}\left(-dt^{\alpha }\right)+\frac{\Lambda }{d}\left(-dt^{\alpha }E_{\alpha ,\alpha +1}\left(-dt^{\alpha }\right)+\frac{1}{\Gamma (1)}\right)\\ \; & =\frac{\Lambda }{d\Gamma (1)}=\frac{\Lambda }{d}. \end{array}$$Consequently, inequality (Equation 9) and the last equation of system (Equation 8), implies that $D_{0}^{\alpha }L\left(t\right)\leq \frac{\mu \Lambda }{d}-\phi L\left(t\right).$ Similar to the above discussion, taking the Laplace transform and using the Mittag-Leffler function, it follows that $L\left(t\right)\leq \frac{\mu \Lambda }{\phi d}$ for any t>0.In conclusion, for any X(0)∈Ω we have $N\left(t\right)\leq \frac{\Lambda }{d}$ and $L\left(t\right)\leq \frac{\mu \Lambda }{\phi d}$ for any t≥0, and thus Ω is positively invariant. □

## 3. Equilibrium Points and Basic Reproduction Number

One way to see what will happen to the population eventually is to explore when the system is at equilibrium. By setting
$$D_{0}^{\alpha }S\left(t\right)=D_{0}^{\alpha }S_{L}\left(t\right)=D_{0}^{\alpha }I\left(t\right)=D_{0}^{\alpha }I_{L}\left(t\right)=D_{0}^{\alpha }L\left(t\right)=0$$
we get the following equilibrium points
$$E_{0}=\left(\frac{\Lambda }{d},0,0,0,0\right),\;\;E_{1}=\left(S_{1},0,I_{1},0,0\right),$$
and
$$E_{2}=\left(S^{*},S_{L}^{*},I^{*},I_{L}^{*},L^{*}\right),$$
where
(10)$$S_{1}=\frac{\gamma_{1}+\alpha_{1}+d}{\beta },\;\;I_{1}=\frac{\Lambda \beta -d(\gamma_{1}+\alpha_{1}+d)}{\beta (\alpha_{1}+d)}$$
(11)$$\begin{array}{cc} S^{*} & =\frac{\gamma_{1}+\alpha_{1}+d}{\beta }+\frac{\lambda_{2}\mu \left(d+\gamma_{2}+\alpha_{2}\right)I^{*}}{\beta \phi (d+\theta_{2}+\gamma_{2}+\alpha_{2})},\;S_{L}^{*}=\frac{\lambda \mu S^{*}I^{*}}{\phi (d+\theta_{1})}\\ I_{L}^{*} & =\frac{\lambda_{2}\mu {\left(I^{*}\right)}^{2}}{\phi (d+\theta_{2}+\alpha_{2}+\gamma_{2})},\;L^{*}=\frac{\mu I^{*}}{\phi } \end{array}$$
and $I^{*}=\max\left\{\tilde{I},0\right\}$, where $\tilde{I}$ is the solution of the equation $k_{1}{\tilde{I}}^{2}+k_{2}\tilde{I}+k_{3}=0$, where
$$\begin{array}{cc} k_{1} & =-\frac{\mu (\lambda_{2}-\theta_{2})}{\phi }\left[1-\frac{\gamma_{2}}{\alpha_{2}+\gamma_{2}+d}-\frac{\theta_{1}\lambda_{1}}{\beta \phi (d+\theta_{1})}+\frac{\mu \lambda_{1}}{\beta \phi }\right]\\ k_{2} & =-\left(\gamma_{1}+\alpha_{1}+d\right)\left[1+\frac{\lambda_{1}\mu }{\beta \phi }-\frac{\gamma_{1}}{\left(\alpha_{1}+\gamma_{1}+d\right)\left(d+\theta_{1}\right)\beta \phi }+\frac{\mu d(\lambda_{2}-\theta_{2})}{(\alpha_{1}+\gamma_{1}+d)\beta \phi }\right]\;\mathrm{and}\\ k_{3} & =\Lambda -\frac{d(\gamma_{1}+\alpha_{1}+d)}{\beta }. \end{array}$$

The disease-free equilibrium $E_{0}$ is the case when the pathogen has suffered extinction and, in the long run, everyone in the population is susceptible. The endemic equilibrium $E_{1}$ is the state where the disease cannot be totally eradicated and remains in the population without a lockdown, while $E_{2}$ is the endemic equilibrium point in the presence of lockdown. The basic reproduction number of the model, denoted by $R_{0}$, is a constant that is used to approximate the expected number of cases directly generated by one case in a population where all individuals are susceptible to infection. One way to calculate $R_{0}$ is using the next generation matrix approach [35]. The rate at which secondary infections are produced is given by
$$F\left(E_{0}\right):=\left(\frac{\partial F_{i}}{\partial x_{j}}\right)\left(E_{0}\right)=\left[\begin{array}{cc} \frac{\beta \Lambda }{d} & \theta_{2}\\ 0 & 0 \end{array}\right],$$
where i,j∈{1,2} and $x_{j}\in \left\{I,I_{L}\right\}$ for j=1,2.

Similarly, the transfer of infection from compartment *i* to *j* is given by
$$V\left(E_{0}\right):=\left(\frac{\partial V_{i}}{\partial x_{j}}\right)\left(E_{0}\right)=\left[\begin{array}{cc} \gamma_{1}+\alpha_{1}+d & 0\\ 0 & d+\theta_{2}+\gamma_{2}+\alpha_{2} \end{array}\right],$$
where i,j∈{1,2} and $x_{j}\in \left\{I,I_{L}\right\}$ for j=1,2.

Now define the next generation matrix *G* as
$$G=FV^{-1}=\left[\begin{array}{cc} \frac{\beta \Lambda }{d(\gamma_{1}+\alpha_{1}+d)} & \frac{\theta_{2}}{d+\theta_{2}+\gamma_{2}+\alpha_{2}}\\ 0 & 0 \end{array}\right].$$

Hence $R_{0}$ is the dominant eigenvalue of *G* and it is given by
(12)$$R_{0}=\frac{\beta \Lambda }{d(\gamma_{1}+\alpha_{1}+d)}.$$

## 4. Local and Global Asymptotic Stability of the Disease-Free and Endemic Equilibrium Points

We use the following theorem to prove the local asymptotic stability of the disease-free equilibrium point $E_{0}.$

**Theorem** **3**([36]). *Given a fractional-order system of differential equation*
(13)$$D_{0}^{\alpha }x\left(t\right)=f\left(x\right),\;0<\alpha \leq 1.$$
*Let $x_{0}$ be an equilibrium point of the given system, and let $A=D(f\left(x_{0}\right))$ be the Jacobian matrix of f evaluated at $x_{0}$. Then $x_{0}$ is locally asymptotically stable if and only if $|arg\left(\lambda_{i}\right)|>\frac{\alpha \pi }{2}$, for all eigenvalues $\lambda_{i}$ of the matrix A.*


**Theorem** **4.**
*The disease-free equilibrium point $E_{0}$ is locally asymptotically stable if $R_{0}<1.$*


**Proof.** After calculating the associated Jacobian matrix of system (Equation 8), it can be shown that the characteristic equation of the Jacobian matrix satisfies
$$\left(\lambda +d\right)\left(\lambda +\phi \right)\left(\lambda +d+\theta_{1}\right)\left(\lambda +d+\theta_{2}+\gamma_{2}+\alpha_{2}\right)\left(\lambda +\left(-\frac{\beta \Lambda }{d}+\gamma_{1}+\alpha_{1}+d\right)\right)=0.$$Note that $R_{0}<1$ iff $-\frac{\beta \Lambda }{d}+\gamma_{1}+\alpha_{1}+d>0.$ As a result, all the eigenvalues are negative real numbers, and hence $\left|arg\left(\lambda \right)\right|=\pi >\frac{\alpha \pi }{2}$ for any 0<α≤1. Thus by Theorem 3 the disease-free equilibrium point $E_{0}$ is locally asymptotically stable. □

**Definition** **7**([32,37]). *An equilibrium point $x^{*}$ of system (Equation 8) is Mittag-Leffler stable if*
$$||x\left(t\right)-x^{*}\left|\right|\leq {\left\{m\left(x_{0}-x^{*}\right)E_{\alpha }\left(-\lambda t^{\alpha }\right)\right\}}^{b},$$*where ||.|| is any norm on $R^{3},\lambda >0,b>0,m\left(0\right)=0$ and m(x)≥0, where m(x) is locally Lipschitz on* Ω* with Lipschitz constant $m_{0}$, and $E_{\alpha }\left(\cdot \right)$ is a one-parameter Mittag-Leffler function which can be defined in terms of the two-parameter Mittag-Leffler function as $E_{\alpha }\left(\cdot \right)=E_{\alpha ,1}\left(\cdot \right)$.*

**Remark** **5.**
*Since Mittage-Leffler stability implies global asymptotic stability (see Remark 4.4 in [38]), it is sufficient to prove the disease-free equilibrium point $E_{0}$ is Mittage-Leffler stable on Ω.*

*Note that a similar result is provided in Definition 5 [39] for a generalized Hattaf fractional (GHF) derivative which encloses the popular forms fractional derivatives with non-singular kernels.*


**Theorem** **5.**
*If $R_{0}<1$, then the disease-free equilibrium point $E_{0}$ is Mittage-Leffler stable on Ω.*


**Proof.** Define the positive definite Lyapunov function
(14)$$V\left(t\right)=c_{1}\left(\frac{\Lambda }{d}-S\left(t\right)\right)+S_{L}\left(t\right)+I\left(t\right)+I_{L}\left(t\right)+c_{2}L\left(t\right),$$
where $c_{1}$ and $c_{2}$ are two positive constants that will be determined later in the proof. By linearity of the Caputo fractional derivative, we have
(15)$$\begin{array}{cc} D_{0}^{\alpha }V\left(t\right) & =-c_{1}D_{0}^{\alpha }S\left(t\right)+D_{0}^{\alpha }\left(S_{L}\left(t\right)+I\left(t\right)+I_{L}\left(t\right)\right)+c_{2}D_{0}^{\alpha }L\left(t\right)\\ \; & =-c_{1}\left(\Lambda -\beta S\left(t\right)I\left(t\right)-\lambda_{1}S\left(t\right)L\left(t\right)-dS\left(t\right)+\gamma_{1}I\left(t\right)+\gamma_{2}I_{L}\left(t\right)+\theta_{1}S_{L}\left(t\right)\right)\\ \; & +(\lambda_{1}S\left(t\right)L\left(t\right)-dS_{L}\left(t\right)-\theta_{1}S_{L}\left(t\right)+\beta S\left(t\right)I\left(t\right)-\gamma_{1}I\left(t\right))\\ \; & -\left(\gamma_{1}+\alpha_{1}+d\right)I\left(t\right)-\left(d+\gamma_{2}+\alpha_{2}\right)I_{L}\left(t\right)\\ \; & =-c_{1}\left(\frac{\Lambda }{d}-S\left(t\right)\right)-\left(c_{1}\theta +\theta_{1}+d\right)S_{L}\left(t\right)-\left(c_{1}\gamma_{1}+\gamma_{1}+\alpha_{1}+d-c_{2}\mu \right)I\left(t\right)\\ \; & -\left(c_{1}\gamma_{2}+d+\gamma_{2}+\alpha_{2}\right)I_{L}\left(t\right)-c_{2}\phi L\left(t\right)+c_{1}\beta S\left(t\right)I\left(t\right)+c_{1}\lambda_{1}S\left(t\right)L\left(t\right)\\ \; & +\lambda_{1}S\left(t\right)L\left(t\right)+\beta S\left(t\right)I\left(t\right). \end{array}$$Now since
$$\begin{array}{cc} c_{1}\beta S\left(t\right)I\left(t\right)+c_{1}\lambda_{1}S\left(t\right)L\left(t\right)+\lambda_{1}S\left(t\right)L\left(t\right)+\beta S\left(t\right)I\left(t\right) & =\left(1+c_{1}\right)\left(\beta S\left(t\right)I\left(t\right)+\lambda_{1}S\left(t\right)Lt\right))\\ \; & \leq \left(1+c_{1}\right)\frac{\Lambda }{d}\left(\beta I\left(t\right)+\lambda_{1}L\left(t\right)\right), \end{array}$$
then Equation (Equation 15) can be rewritten as
(16)$$\begin{array}{cc} D_{0}^{\alpha }V\left(t\right) & \leq -c_{1}\left(\frac{\Lambda }{d}-S\left(t\right)\right)-\left(c_{1}\theta +\theta_{1}+d\right)S_{L}\left(t\right)\\ \; & -\left(\gamma_{1}+\alpha_{1}+d-c_{2}\mu -\frac{\beta \Lambda }{d}\left(1+c_{1}\right)\right)I\left(t\right)-\left(c_{1}\gamma_{2}+d+\gamma_{2}+\alpha_{2}\right)I_{L}\left(t\right)\\ \; & -\left(c_{2}\phi -\left(c_{1}+1\right)\lambda_{1}\frac{\Lambda }{d}\right)L\left(t\right). \end{array}$$If $R_{0}=\frac{\beta \Lambda }{d(\gamma_{1}+\alpha_{1}+d)}<1$ then we can choose $c_{1},c_{2}>0$ such that
$$\gamma_{1}+\alpha_{1}+d-c_{2}\mu -\frac{\beta \Lambda }{d}\left(1+c_{1}\right)>0,\;c_{2}\phi -\left(c_{1}+1\right)\lambda_{1}\frac{\Lambda }{d}>0.$$Thus the inequality in (Equation 16) becomes
(17)$$D_{0}^{\alpha }V\left(t\right)\leq -c_{3}V\left(t\right)$$
where
$$\begin{array}{ccc} c_{3} & = & \min\left\{c_{1},\;c_{1}\theta +\theta_{1}+d,\;\gamma_{1}+\alpha_{1}+d-c_{2}\mu -\frac{\beta \Lambda }{d}\left(1+c_{1}\right),\;c_{1}\gamma_{2}+d+\gamma_{2}+\alpha_{2},\right.\\ & & \left.c_{1}\phi -\left(c_{1}+1\right)\frac{\lambda_{1}\Lambda }{d}\right\}>0. \end{array}$$Now taking the Laplace transform of inequality (Equation 17) results
(18)$$LV\left(t\right)\leq \frac{s^{\alpha -1}}{s^{\alpha }+c_{3}}V\left(0\right).$$Thus using (2), we have
(19)$$V\left(t\right)\leq L^{-1}\left(\frac{s^{\alpha -1}V\left(0\right)}{s^{\alpha }+c_{3}}\right)=V\left(0\right)E_{\alpha }\left(-c_{3}t^{\alpha }\right),\;\forall t\geq 0.$$Let $\left||x\left(t\right)|\right|=\left|S\left(t\right)|+\right|S_{L}\left(t\right)|+|I\left(t\right)|+|I_{L}\left(t\right)|+|L\left(t\right)|$ be the norm defined on the solution $x\left(t\right)=(S\left(t\right),S_{L}\left(t\right),I\left(t\right),I_{L}\left(t\right),L\left(t\right))$ of system (Equation 8). Then from Equation (Equation 14) it follows that
(20)$$c_{4}||x\left(t\right)-E_{0}\left|\right|\leq V\left(t\right)\leq c_{5}||x\left(t\right)-E_{0}\left|\right|,\;\forall t\geq 0$$
where $c_{4}=\min\left\{1,c_{1},c_{2}\right\}$ and $c_{5}=\max\left\{1,c_{1},c_{2}\right\}.$ As a result, from inequality (Equation 19) and (Equation 20) it follows that $||x\left(t\right)-E_{0}\left|\right|\leq M\left(x_{0}-E_{0}\right)E_{\alpha }\left(-c_{3}t^{\alpha }\right),$ where $M\left(S,S_{L},I,I_{L},L\right):=\frac{c_{4}}{c_{5}}\left(S\left(t\right)+S_{L}\left(t\right)+I\left(t\right)+I_{L}\left(t\right)+L\left(t\right)\right)$. In conclusion, the disease-free equilibrium point is Mittage-Leffler stable, and thus it is globally asymptotically stable. □

**Remark** **6.**
*The proof of the local stability of the endemic equilibrium point $E_{1}$ is similar to Theorem 4. Thus we will provide proof of the global stability of $E_{1}$ by constructing a Lyapunov function.*


**Theorem** **6**([40]). *Let $x_{0}\in \Gamma$ be an equilibrium point for the non-autonomous fractional-order system $D_{0}^{\alpha }x\left(t\right)=f\left(t,x\right)$. Also let L:[0,∞)×Γ→R be a continuously differentiable function such that*
$$W_{1}\left(x\right)\leq L\left(t,x\left(t\right)\right)\leq W_{2}\left(x\right)$$
*and*
$$D_{0}^{\alpha }L\left(t,x\left(t\right)\right))\leq -W_{3}\left(x\right)$$*for all α∈(0,1) and all x∈Γ, where $W_{1},\;W_{2},\;W_{3}$ are continuous positive definite functions on Γ. Then the equilibrium point $x_{0}$ is globally asymptotically stable.*

Now using Theorem 6 we will show that the endemic equilibrium point $E_{1}$ of system (Equation 8) is globally asymptotically stable under some conditions.

**Theorem** **7.***If $R_{0}>1$, then $E_{1}$ is globally asymptotically stable in* Ω.

**Proof.** Consider the Lyapunov function
(21)$$V\left(S,S_{L},I,I_{L},L\right)=L_{1}\left(S\right)+L_{2}\left(I\right)+L_{3}\left(S_{L},I_{L},L\right),$$
where
$$\begin{array}{cc} L_{1} & =S\left(t\right)-S_{1}-S_{1}\ln\left(\frac{S(t)}{S_{1}}\right),\;L_{2}=I\left(t\right)-I_{1}-I_{1}\ln\left(\frac{I(t)}{I_{1}}\right)\;\mathrm{and}\\ L_{3} & =S_{L}\left(t\right)+I_{L}\left(t\right)+kL\left(t\right), \end{array}$$
and *k* is a positive constant to be determined later in the proof. Using Lemma 3.1 and Remark 3.1 of [41] we have
(22)$$\begin{array}{cc} D_{0}^{\alpha }L_{1} & =D_{0}^{\alpha }\left[S\left(t\right)-S_{1}-S_{1}\ln\left(\frac{S(t)}{S_{1}}\right)\right]\leq \left(1-\frac{S_{1}}{S(t)}\right)D_{0}^{\alpha }S\left(t\right) \end{array}$$
(23)$$\begin{array}{cc} D_{0}^{\alpha }L_{2} & =D_{0}^{\alpha }\left[I\left(t\right)-I_{1}-I_{1}\ln\left(\frac{I(t)}{I_{1}}\right)\right]\leq \left(1-\frac{I_{1}}{I(t)}\right)D_{0}^{\alpha }I\left(t\right), \end{array}$$
and by the linearity property of the fractional Caputo derivative, it follows that
(24)$$D_{0}^{\alpha }L_{3}\left(t\right)=D_{0}^{\alpha }S_{L}\left(t\right)+D_{0}^{\alpha }I_{L}\left(t\right)+kD_{0}^{\alpha }L\left(t\right).$$Now using Equations (Equation 10), (Equation 22)–(Equation 24) and the fact that
$$D_{0}^{\alpha }V=D_{0}^{\alpha }L_{1}+D_{0}^{\alpha }L_{2}+D_{0}^{\alpha }L_{3},$$
we have
(25)$$\begin{array}{cc} D_{0}^{\alpha }V & \leq \left(1-\frac{S_{1}}{S(t)}\right)D_{0}^{\alpha }S\left(t\right)+\left(1-\frac{I_{1}}{I(t)}\right)D_{0}^{\alpha }I\left(t\right)+D_{0}^{\alpha }S_{L}\left(t\right)+D_{0}^{\alpha }I_{L}\left(t\right)+kD_{0}^{\alpha }L\left(t\right)\\ \; & \leq \frac{-d-\beta I_{1}}{S}{\left(S-S_{1}\right)}^{2}+\left(\gamma_{1}+\alpha_{1}+d-\frac{\beta \Lambda }{d}\right){\left(I-I_{1}\right)}^{2}\\ \; & +\left(S_{1}\lambda_{1}-k\phi \right)L-dS_{L}. \end{array}$$Choose any $k>\frac{S_{1}\lambda_{2}}{\phi }$ and note that if $R_{0}>1$ then $\gamma_{1}+\alpha_{1}+d-\frac{\beta \Lambda }{d}<0.$ Thus if $R_{0}>1$ then $D_{0}^{\alpha }V\leq 0$ and also $D_{0}^{\alpha }V=0$ if and only if $S=S_{1}$, $I=I_{1}$ and $S_{L}=I_{L}=L=0.$ Therefore the largest compact invariant set in
$$E=\left\{\left(S,S_{L},I,I_{L},L\right)\in \Omega :D_{0}^{\alpha }V=0\right\}$$
is the singleton set containing the endemic equilibrium $E_{1}.$ Thus by Theorem 6, we conclude that the endemic equilibrium is globally asymptotically stable in Ω. □

Since the proof of the local stability of the endemic equilibrium point with lockdown $E_{2}$ is similar to the proof of (4), we will provide the proof of the global stability of the endemic equilibrium point with lockdown $E_{2}.$

**Theorem** **8.***If $\mu \lambda_{2}I^{*}-\mu \theta_{2}I^{*}-\alpha_{2}\phi L^{*}-\phi^{2}<0$, then the endemic equilibrium point with lockdown, $E_{2}$, is globally asymptotically stable in* Ω.

**Proof.** We start by defining a Lyapunov function
$$\begin{array}{cc} V= & \left(S-S^{*}-S^{*}\ln\left(S\right)\right)+\left(S_{L}-S_{L}^{*}-S_{L}^{*}\ln\left(S_{L}\right)\right)+\left(I-I^{*}-I^{*}\ln\left(I\right)\right)\\ \; & \left(I_{L}-I_{L}^{*}-I_{L}^{*}\ln\left(I_{L}\right)\right)+\left(L-L^{*}-L^{*}\ln\left(L\right)\right), \end{array}$$
where $E_{2}=\left(S^{*},S_{L}^{*},I^{*},I_{L}^{*},L^{*}\right)$ is the endemic equilibrium point with lockdown given in Equation (Equation 11). Note that $V(E_{2})=0$ and using Lemma 3.1 and Remark 3.1 of [41] we have the following
$$\begin{array}{cc} D_{0}^{\alpha }V & \leq \left(1-\frac{S^{*}}{S}\right)D_{0}^{\alpha }S\left(t\right)+\left(1-\frac{S_{L}^{*}}{S_{L}}\right)D_{0}^{\alpha }S_{L}\left(t\right)+\left(1-\frac{I^{*}}{I}\right)D_{0}^{\alpha }I\left(t\right)\\ \; & +\left(1-\frac{I_{L}^{*}}{I_{L}}\right)D_{0}^{\alpha }I_{L}\left(t\right)+\left(1-\frac{L^{*}}{L}\right)D_{0}^{\alpha }L\left(t\right)\\ \; & \leq \left(S-S^{*}\right)\left(\frac{\Lambda }{S}\beta I-d\right)+S^{*}\lambda_{1}L-d\left(S_{L}-S_{L}^{*}\right)-\frac{S_{L}^{*}\lambda_{1}SL}{S_{L}}+S_{L}^{*}\theta_{1}\\ \; & +\left(I-I^{*}\right)\left(\beta S-\alpha_{1}-d\right)+I^{*}\gamma_{1}+\lambda_{2}I^{*}L-\frac{\theta_{2}I^{*}I_{L}}{I}\\ \; & -\left(I_{L}-I_{L}^{*}\right)\left(d+\alpha_{2}\right)-I_{L}^{*}\left(\frac{\lambda_{2}IL}{I_{L}}-\theta_{2}-\gamma_{2}\right)+\left(L-L^{*}\right)\left(\frac{\mu I}{L}-\phi \right) \end{array}$$Using Equation (Equation 11) we have the following inequality
$$\begin{array}{cc} D_{0}^{\alpha }V & \leq \left(-\beta I^{*}-\frac{\lambda_{1}\mu I^{*}}{\phi }-d\right){\left(S-S^{*}\right)}^{2}-\left(d+\theta \right){\left(S_{L}-S_{L}^{*}\right)}^{2}\\ \; & +\left(\mu \lambda_{2}I^{*}-\mu \theta_{2}I^{*}-\alpha_{2}\phi L^{*}-\phi^{2}\right){\left(I-I^{*}\right)}^{2}\\ \; & -\left(d+\alpha_{2}\right)I_{L}^{2}+\left(d+\alpha_{2}\right)I_{L}^{*}I_{L}-\left(\lambda_{2}IL+d+\alpha_{2}+\theta_{2}+\gamma_{2}\right){\left(I_{L}^{*}\right)}^{2}\\ \; & -\phi^{2}L^{2}+\left(\mu I+\mu I^{*}\right)\phi L+\left(\mu^{2}II^{*}-I^{2}\mu^{2}-\mu^{2}{\left(I^{*}\right)}^{2}\right). \end{array}$$Let $f\left(I_{L}\right):=-\left(d+\alpha_{2}\right)I_{L}^{2}+\left(d+\alpha_{2}\right)I_{L}^{*}I_{L}-\left(\lambda_{2}IL+d+\alpha_{2}+\theta_{2}+\gamma_{2}\right){\left(I_{L}^{*}\right)}^{2}.$ Then it follows that
$${\left(\left(d+\alpha_{2}\right)I_{L}^{*}\right)}^{2}-4\left(\left(d+\alpha_{2}\right)\right)\left(\left(\lambda_{2}IL+d+\alpha_{2}+\theta_{2}+\gamma_{2}\right){\left(I_{L}^{*}\right)}^{2}\right)<0.$$Thus it follows that $f(I_{L})<0$. Similarly, it can be shown that if$g\left(L\right):=-\phi^{2}L^{2}+\left(\mu I+\mu I^{*}\right)\phi L+\left(\mu^{2}II^{*}-I^{2}\mu^{2}-\mu^{2}{\left(I^{*}\right)}^{2}\right)$, then g(L)<0 since
$${\left(\left(\mu I+\mu I^{*}\right)\phi \right)}^{2}-4\left(\phi^{2}\right)\left(\mu^{2}II^{*}-I^{2}\mu^{2}-\mu^{2}{\left(I^{*}\right)}^{2}\right)<0$$
and the fact that $(\mu^{2}II^{*}-I^{2}\mu^{2}-\mu^{2}{\left(I^{*}\right)}^{2})<0$. Thus under the given assumption that $\mu \lambda_{2}I^{*}-\mu \theta_{2}I^{*}-\alpha_{2}\phi L^{*}-\phi^{2}<0$ it can be concluded that $D_{0}^{\alpha }V\leq 0$ and $D_{0}^{\alpha }V=0$ if and only if $S=S^{*},S_{L}=S_{L}^{*},I=I^{*}$. This result, together with the last two equations of (Equation 8), will imply that $I_{L}=I_{L}^{*}$ and $L=L^{*}.$ As a result, the endemic equilibrium point $E_{2}$ is globally asymptotically stable. □

## 5. Approximate Solution

In this section, we present a solution of the fractional-order model by using the residual power series method, which consists of expressing the solution as fractional power series expanded about the initial point $t=t_{0}$.

Consider the following fractional-order model:(26)$$\begin{array}{c} D_{0}^{\alpha }S\left(t\right)=\Lambda -\beta SI-\lambda_{1}SL-dS+\gamma_{1}I+\gamma_{2}I_{L}+\theta_{1}S_{L}\\ D_{0}^{\alpha }S_{L}\left(t\right)=\lambda_{1}SL-hS_{L}\\ D_{0}^{\alpha }I\left(t\right)=\beta SI-pI-\lambda_{2}IL+\theta_{2}I_{L}\\ D_{0}^{\alpha }I_{L}\left(t\right)=\lambda_{2}IL-rI_{L}\\ D_{0}^{\alpha }L\left(t\right)=\mu I-\phi L, \end{array}$$
where $D_{0}^{\alpha }$ denotes the Caputo derivative for 0<α≤1 with $h=d+\theta_{1},p=\gamma_{1}+\alpha_{1}+d$ and $r=d+\theta_{2}+\gamma_{2}+\alpha_{2}$; under the initial conditions
$$S\left(0\right)=S_{0}>0,\;\;S_{L}\left(0\right)=S_{L}^{0}>0,\;\;I\left(0\right)=I^{0}>0,\;\;I_{L}\left(0\right)=I_{L}^{0}>0\;\mathrm{and}\;L\left(0\right)=L_{0}>0.$$

Similar to the procedure in [42], we do the following steps in order to obtain a fractional power series solution for the nonlinear fractional-order model in (Equation 26).

**Step 1:** Suppose that the fractional power series of $S\left(t\right),S_{L}\left(t\right),I\left(t\right),I_{L}\left(t\right)$ and L(t) around t=0 are as follows:(27)$$\begin{array}{cc} S(t) & =\sum_{k=0}^{\infty }\frac{a_{k}}{\Gamma (1+k\alpha )}t^{k\alpha },\\ S_{L}\left(t\right) & =\sum_{k=0}^{\infty }\frac{b_{k}}{\Gamma (1+k\alpha )}t^{k\alpha },\\ I(t) & =\sum_{k=0}^{\infty }\frac{c_{k}}{\Gamma (1+k\alpha )}t^{k\alpha },\\ I_{L}\left(t\right) & =\sum_{k=0}^{\infty }\frac{d_{k}}{\Gamma (1+k\alpha )}t^{k\alpha },\\ L(t) & =\sum_{k=0}^{\infty }\frac{e_{k}}{\Gamma (1+k\alpha )}t^{k\alpha }, \end{array}$$
where 0≤t<η for some η>0. Now, we let $S_{n}\left(t\right),S_{L,n}\left(t\right),I_{n}\left(t\right),I_{L,n}\left(t\right)$ and $L_{n}\left(t\right)$ denote the *n*-th truncated power series approximation of $S\left(t\right),S_{L}\left(t\right),I\left(t\right),I_{L}\left(t\right)$ and L(t), respectively. That is,
(28)$$\begin{array}{cc} S_{n}\left(t\right) & =\sum_{k=0}^{n}\frac{a_{k}}{\Gamma (1+k\alpha )}t^{k\alpha },\\ S_{L,n}\left(t\right) & =\sum_{k=0}^{n}\frac{b_{k}}{\Gamma (1+k\alpha )}t^{k\alpha },\\ I_{n}\left(t\right) & =\sum_{k=0}^{n}\frac{c_{k}}{\Gamma (1+k\alpha )}t^{k\alpha },\\ I_{L,n}\left(t\right) & =\sum_{k=0}^{n}\frac{d_{k}}{\Gamma (1+k\alpha )}t^{k\alpha }\\ L_{n}\left(t\right) & =\sum_{k=0}^{n}\frac{e_{k}}{\Gamma (1+k\alpha )}t^{k\alpha }. \end{array}$$

**Step 2:** Next, we define the residual functions for the model in (Equation 26) as follows:(29)$$\begin{array}{cc} Res_{S}\left(t\right) & =D_{0}^{\alpha }S\left(t\right)-\Lambda +\beta S\left(t\right)I\left(t\right)+\lambda_{1}S\left(t\right)L\left(t\right)+dS\left(t\right)-\gamma_{1}I\left(t\right)\\ \; & \;\;\;\;\;\;\;\;\;\;\;\;\;\;\;\;\;\;\;\;\;\;\;\;\;\;\;\;\;\;\;\;\;\;\;-\gamma_{2}I_{L}\left(t\right)-\theta_{1}S_{L}\left(t\right)\\ Res_{S_{L}}\left(t\right) & =D_{0}^{\alpha }S_{L}\left(t\right)-\lambda_{1}S\left(t\right)L\left(t\right)+hS_{L}\left(t\right)\\ Res_{I}\left(t\right) & =D_{0}^{\alpha }I\left(t\right)-\beta S\left(t\right)I\left(t\right)+pI\left(t\right)+\lambda_{2}I\left(t\right)L\left(t\right)-\theta_{2}I_{L}\left(t\right)\\ Res_{I_{L}}\left(t\right) & =D_{0}^{\alpha }I_{L}\left(t\right)-\lambda_{2}I\left(t\right)L\left(t\right)+rI_{L}\left(t\right)\\ Res_{L}\left(t\right) & =D_{0}^{\alpha }L\left(t\right)-\mu I\left(t\right)+\phi L\left(t\right). \end{array}$$

Hence, the *n*-th residual functions for $S\left(t\right),S_{L}\left(t\right),I\left(t\right),I_{L}\left(t\right)$ and L(t), respectively, are as follows:(30)$$\begin{array}{cc} Res_{S_{n}}\left(t\right) & =D_{0}^{\alpha }S_{n}\left(t\right)-\Lambda +\beta S_{n}\left(t\right)I_{n}\left(t\right)+\lambda_{1}S_{n}\left(t\right)L_{n}\left(t\right)+dS_{n}\left(t\right)-\gamma_{1}I_{n}\left(t\right)\\ \; & \;\;\;\;\;\;\;\;\;\;\;\;\;\;\;\;\;\;\;\;\;\;\;\;\;\;\;\;\;\;\;\;\;\;\;-\gamma_{2}I_{L,n}\left(t\right)-\theta_{1}S_{L,n}\left(t\right)\\ Res_{S_{L,n}}\left(t\right) & =D_{0}^{\alpha }S_{L,n}\left(t\right)-\lambda_{1}S_{n}\left(t\right)L_{n}\left(t\right)+hS_{L,n}\left(t\right)\\ Res_{I_{n}}\left(t\right) & =D_{0}^{\alpha }I_{n}\left(t\right)-\beta S_{n}\left(t\right)I_{n}\left(t\right)+pI_{n}\left(t\right)+\lambda_{2}I_{n}\left(t\right)L_{n}\left(t\right)-\theta_{2}I_{L,n}\left(t\right)\\ Res_{I_{L,n}}\left(t\right) & =D_{0}^{\alpha }I_{L,n}\left(t\right)-\lambda_{2}I_{n}\left(t\right)L_{n}\left(t\right)+rI_{L,n}\left(t\right)\\ Res_{L_{n}}\left(t\right) & =D_{0}^{\alpha }L_{n}\left(t\right)-\mu I_{n}\left(t\right)+\phi L_{n}\left(t\right). \end{array}$$

Now, we observe that

$Res_{S_{n}}\left(t\right)=Res_{S_{L,n}}\left(t\right)=Res_{I_{n}}\left(t\right)=Res_{I_{L,n}}\left(t\right)=Res_{L_{n}}\left(t\right)=0$, and
$$\lim_{n\to \infty }Res_{S_{n}}\left(t\right)=Res_{S}\left(t\right),\lim_{n\to \infty }Res_{S_{L,n}}\left(t\right)=Res_{S_{L}}\left(t\right),\lim_{n\to \infty }Res_{I_{n}}\left(t\right)=Res_{I}\left(t\right),$$$$\lim_{n\to \infty }Res_{I_{L,n}}\left(t\right)=Res_{I_{L}}\left(t\right)\;\mathrm{and}\;\lim_{n\to \infty }Res_{L_{n}}\left(t\right)=Res_{L}\left(t\right)\;\mathrm{for}\;\mathrm{all}\;t\geq 0.$$

Since the Caputo derivative of any constant is zero, it is straightforward to see that for k=1,⋯,n
$$\begin{array}{cc} \; & D_{0}^{(k-1)\alpha }Res_{S}\left(0\right)=D_{0}^{(k-1)\alpha }Res_{S_{n}}\left(0\right),\;\;D_{0}^{(k-1)\alpha }Res_{S_{L}}\left(0\right)=D_{0}^{(k-1)\alpha }Res_{S_{L,n}}\left(0\right)\\ \; & D_{0}^{(k-1)\alpha }Res_{I}\left(0\right)=D_{0}^{(k-1)\alpha }Res_{I_{n}}\left(0\right),\;\;D_{0}^{(k-1)\alpha }Res_{I_{L}}\left(0\right)=D_{0}^{(k-1)\alpha }Res_{I_{L,n}}\left(0\right)\\ \; & \mathrm{and}\;\;D_{0}^{(k-1)\alpha }Res_{L}\left(0\right)=D_{0}^{(k-1)\alpha }Res_{L_{n}}\left(0\right). \end{array}$$

**Step 3:** To obtain the coefficients $a_{k},b_{k},c_{k},d_{k}$ and $e_{k}$, for k=1,⋯,n, we substitute the *n*-th truncated series of $S\left(t\right),S_{L}\left(t\right),I\left(t\right),I_{L}\left(t\right)$ and L(t) into (Equation 30) and then apply the Caputo fractional derivative operator $D_{0}^{(n-1)\alpha }$ on $S_{n}\left(t\right),S_{L,n}\left(t\right),I_{n}\left(t\right),I_{L,n}\left(t\right)$ and $L_{n}\left(t\right)$, and evaluate the result at t=0. We obtain the following algebraic system of equations.
(31)$$\begin{array}{cc} D_{0}^{(n-1)\alpha }Res_{S_{n}}\left(0\right) & =0,\\ D_{0}^{(n-1)\alpha }Res_{S_{L,n}}\left(0\right) & =0,\\ D_{0}^{(n-1)\alpha }Res_{I_{n}}\left(0\right) & =0\\ D_{0}^{(n-1)\alpha }Res_{I_{L,n}}\left(0\right) & =0,\\ D_{0}^{(n-1)\alpha }Res_{L_{n}}\left(0\right) & =0. \end{array}$$
for n=1,2,3,⋯.

**Step 4:** We solve the algebraic system (Equation 31) for the values of $a_{k},b_{k},c_{k},d_{k},e_{k}$ for k=1,2,3,⋯,n to get the *n*-th residual power series approximate solution of the system in (Equation 26).

**Step 5:** We repeat the procedure to obtain a sufficient number of coefficients. Higher accuracy for the solution can be achieved by evaluating more terms in the series solution.

By following the steps outlined in above, we derive a recursive formula for the coefficients as follows:

First, we note that the coefficients $a_{0},b_{0},c_{0},d_{0}$ and $e_{0}$ are given by the initial conditions. That is,
$$\begin{array}{c} a_{0}=S_{0},\;\;b_{0}=S_{L}^{0},\;\;c_{0}=I^{0},\;\;d_{0}=I_{L}^{0}\;\;\mathrm{and}\;\;e_{0}=L_{0}. \end{array}$$

The first truncated power series approximations will have the forms:$$\begin{array}{cc} S_{1}\left(t\right) & =a_{0}+\frac{a_{1}}{\Gamma (1+\alpha )}t^{\alpha },\\ S_{L,1}\left(t\right) & =b_{0}+\frac{b_{1}}{\Gamma (1+\alpha )}t^{\alpha },\\ I_{1}\left(t\right) & =c_{0}+\frac{c_{1}}{\Gamma (1+\alpha )}t^{\alpha },\\ I_{L,1}\left(t\right) & =d_{0}+\frac{d_{1}}{\Gamma (1+\alpha )}t^{\alpha },\;\;\mathrm{and}\\ L_{1}\left(t\right) & =e_{0}+\frac{e_{1}}{\Gamma (1+\alpha )}t^{\alpha }. \end{array}$$

Thus, the first residual functions are:$$\begin{array}{cc} Res_{S_{1}}\left(t\right) & =D_{0}^{\alpha }\left(a_{0}+\frac{a_{1}}{\Gamma (1+\alpha )}t^{\alpha }\right)-\Lambda +\beta \left(a_{0}+\frac{a_{1}}{\Gamma (1+\alpha )}t^{\alpha }\right)\left(c_{0}+\frac{c_{1}}{\Gamma (1+\alpha )}t^{\alpha }\right)\\ \; & +\lambda_{1}\left(a_{0}+\frac{a_{1}}{\Gamma (1+\alpha )}t^{\alpha }\right)\left(e_{0}+\frac{e_{1}}{\Gamma (1+\alpha )}t^{\alpha }\right)\\ \; & +d\left(a_{0}+\frac{a_{1}}{\Gamma (1+\alpha )}t^{\alpha }\right)-\gamma_{1}\left(c_{0}+\frac{c_{1}}{\Gamma (1+\alpha )}t^{\alpha }\right)\\ \; & -\gamma_{2}\left(d_{0}+\frac{d_{1}}{\Gamma (1+\alpha )}t^{\alpha }\right)-\theta_{1}\left(b_{0}+\frac{b_{1}}{\Gamma (1+\alpha )}t^{\alpha }\right),\\ Res_{S_{L,1}}\left(t\right) & =D_{0}^{\alpha }\left(b_{0}+\frac{b_{1}}{\Gamma (1+\alpha )}t^{\alpha }\right)-\lambda_{1}\left(a_{0}+\frac{a_{1}}{\Gamma (1+\alpha )}t^{\alpha }\right)\left(e_{0}+\frac{e_{1}}{\Gamma (1+\alpha )}t^{\alpha }\right)\\ \; & +h\left(b_{0}+\frac{b_{1}}{\Gamma (1+\alpha )}t^{\alpha }\right),\\ Res_{I_{1}}\left(t\right) & =D_{0}^{\alpha }\left(c_{0}+\frac{c_{1}}{\Gamma (1+\alpha )}t^{\alpha }\right)-\beta \left(a_{0}+\frac{a_{1}}{\Gamma (1+\alpha )}t^{\alpha }\right)\left(c_{0}+\frac{c_{1}}{\Gamma (1+\alpha )}t^{\alpha }\right)\\ \; & \;\;\;\;+p\left(c_{0}+\frac{c_{1}}{\Gamma (1+\alpha )}t^{\alpha }\right)+\lambda_{2}\left(c_{0}+\frac{c_{1}}{\Gamma (1+\alpha )}t^{\alpha }\right)\left(e_{0}+\frac{e_{1}}{\Gamma (1+\alpha )}t^{\alpha }\right)\\ \; & \;\;\;\;-\theta_{2}\left(d_{0}+\frac{d_{1}}{\Gamma (1+\alpha )}t^{\alpha }\right),\\ Res_{I_{L,1}}\left(t\right) & =D_{0}^{\alpha }\left(d_{0}+\frac{d_{1}}{\Gamma (1+\alpha )}t^{\alpha }\right)-\lambda_{2}\left(c_{0}+\frac{c_{1}}{\Gamma (1+\alpha )}t^{\alpha }\right)\left(e_{0}+\frac{e_{1}}{\Gamma (1+\alpha )}t^{\alpha }\right)\\ \; & \;\;\;\;+r\left(d_{0}+\frac{d_{1}}{\Gamma (1+\alpha )}t^{\alpha }\right)\;\;\mathrm{and}\\ Res_{L_{1}}\left(t\right) & =D_{0}^{\alpha }\left(e_{0}+\frac{e_{1}}{\Gamma (1+\alpha )}t^{\alpha }\right)-\mu \left(c_{0}+\frac{c_{1}}{\Gamma (1+\alpha )}t^{\alpha }\right)+\phi \left(e_{0}+\frac{e_{1}}{\Gamma (1+\alpha )}t^{\alpha }\right). \end{array}$$

Evaluating $Res_{S_{1}}\left(t\right),Res_{S_{L,1}}\left(t\right),Res_{I_{1}}\left(t\right),Res_{I_{L,1}}\left(t\right)$ and $Res_{L_{1}}\left(t\right)$ at t=0, we have
$$\begin{array}{cc} Res_{S_{1}}\left(0\right) & =a_{1}-\Lambda +\beta a_{0}c_{0}+\lambda_{1}a_{0}e_{0}+da_{0}-\gamma_{1}c_{0}-\gamma_{2}d_{0}-\theta_{1}b_{0},\\ Res_{S_{L,1}}\left(0\right) & =b_{1}-\lambda_{1}a_{0}e_{0}+hb_{0},\\ Res_{I_{1}}\left(0\right) & =c_{1}-\beta a_{0}c_{0}+pc_{0}+\lambda_{2}c_{0}e_{0}-\theta_{2}d_{0},\\ Res_{I_{L,1}}\left(0\right) & =d_{1}-\lambda_{2}c_{0}e_{0}+rd_{0},\\ Res_{L_{1}}\left(0\right) & =e_{1}-\mu c_{0}+\phi e_{0}. \end{array}$$

By solving the equations
$$\begin{array}{c} Res_{S_{1}}\left(0\right)=0,\;\;Res_{S_{L,1}}\left(0\right)=0,\;\;Res_{I_{1}}\left(0\right)=0,\;\;Res_{I_{L,1}}\left(0\right)=0\;\mathrm{and}\;Res_{L_{1}}\left(0\right)=0, \end{array}$$
we have that
$$\begin{array}{cc} a_{1} & =\Lambda -\beta a_{0}c_{0}-\lambda_{1}a_{0}e_{0}-da_{0}+\gamma_{1}c_{0}+\gamma_{2}d_{0}+\theta_{1}b_{0},\\ b_{1} & =\lambda_{1}a_{0}e_{0}-hb_{0},\\ c_{1} & =\beta a_{0}c_{0}-pc_{0}-\lambda_{2}c_{0}e_{0}+\theta_{2}d_{0},\\ d_{1} & =\lambda_{2}c_{0}e_{0}-rd_{0},\\ e_{1} & =\mu c_{0}-\phi e_{0}. \end{array}$$

Now, the second truncated power series approximations will have the forms:$$\begin{array}{cc} S_{2}\left(t\right) & =a_{0}+\frac{a_{1}}{\Gamma (1+\alpha )}t^{\alpha }+\frac{a_{2}}{\Gamma (1+2\alpha )}t^{2\alpha },\\ S_{L,2}\left(t\right) & =b_{0}+\frac{b_{1}}{\Gamma (1+\alpha )}t^{\alpha }+\frac{b_{2}}{\Gamma (1+2\alpha )}t^{2\alpha },\\ I_{2}\left(t\right) & =c_{0}+\frac{c_{1}}{\Gamma (1+\alpha )}t^{\alpha }+\frac{c_{2}}{\Gamma (1+2\alpha )}t^{2\alpha },\\ I_{L,2}\left(t\right) & =d_{0}+\frac{d_{1}}{\Gamma (1+\alpha )}t^{\alpha }+\frac{d_{2}}{\Gamma (1+2\alpha )}t^{2\alpha },\;\;\mathrm{and}\\ L_{2}\left(t\right) & =e_{0}+\frac{e_{1}}{\Gamma (1+\alpha )}t^{\alpha }+\frac{e_{2}}{\Gamma (1+2\alpha )}t^{2\alpha }. \end{array}$$

So, the second residual functions are:$$\begin{array}{cc} Res_{S_{2}}\left(t\right) & =D_{0}^{\alpha }\left(a_{0}+\frac{a_{1}}{\Gamma (1+\alpha )}t^{\alpha }+\frac{a_{2}}{\Gamma (1+2\alpha )}t^{2\alpha }\right)-\Lambda \\ \; & +\beta \left(a_{0}+\frac{a_{1}}{\Gamma (1+\alpha )}t^{\alpha }+\frac{a_{2}}{\Gamma (1+2\alpha )}t^{2\alpha }\right)\left(c_{0}+\frac{c_{1}}{\Gamma (1+\alpha )}t^{\alpha }+\frac{c_{2}}{\Gamma (1+2\alpha )}t^{2\alpha }\right)\\ \; & +\lambda_{1}\left(a_{0}+\frac{a_{1}}{\Gamma (1+\alpha )}t^{\alpha }+\frac{a_{2}}{\Gamma (1+2\alpha )}t^{2\alpha }\right)\left(e_{0}+\frac{e_{1}}{\Gamma (1+\alpha )}t^{\alpha }+\frac{e_{2}}{\Gamma (1+2\alpha )}t^{2\alpha }\right)\\ \; & +d\left(a_{0}+\frac{a_{1}}{\Gamma (1+\alpha )}t^{\alpha }+\frac{a_{2}}{\Gamma (1+2\alpha )}t^{2\alpha }\right)\\ \; & -\gamma_{1}\left(c_{0}+\frac{c_{1}}{\Gamma (1+\alpha )}t^{\alpha }+\frac{c_{2}}{\Gamma (1+2\alpha )}t^{2\alpha }\right)\\ \; & -\gamma_{2}\left(d_{0}+\frac{d_{1}}{\Gamma (1+\alpha )}t^{\alpha }+\frac{d_{2}}{\Gamma (1+2\alpha )}t^{2\alpha }\right)-\theta_{1}\left(b_{0}+\frac{b_{1}}{\Gamma (1+\alpha )}t^{\alpha }+\frac{b_{2}}{\Gamma (1+2\alpha )}t^{2\alpha }\right),\\ Res_{S_{L,2}}\left(t\right) & =D_{0}^{\alpha }\left(b_{0}+\frac{b_{1}}{\Gamma (1+\alpha )}t^{\alpha }+\frac{b_{2}}{\Gamma (1+2\alpha )}t^{2\alpha }\right)\\ \; & -\lambda_{1}\left(a_{0}+\frac{a_{1}}{\Gamma (1+\alpha )}t^{\alpha }+\frac{a_{2}}{\Gamma (1+2\alpha )}t^{2\alpha }\right)\left(e_{0}+\frac{e_{1}}{\Gamma (1+\alpha )}t^{\alpha }+\frac{e_{2}}{\Gamma (1+2\alpha )}t^{2\alpha }\right)\\ \; & +h\left(b_{0}+\frac{b_{1}}{\Gamma (1+\alpha )}t^{\alpha }+\frac{b_{2}}{\Gamma (1+2\alpha )}t^{2\alpha }\right),\\ Res_{I_{2}}\left(t\right) & =D_{0}^{\alpha }\left(c_{0}+\frac{c_{1}}{\Gamma (1+\alpha )}t^{\alpha }+\frac{c_{2}}{\Gamma (1+2\alpha )}t^{2\alpha }\right)\\ \; & -\beta \left(a_{0}+\frac{a_{1}}{\Gamma (1+\alpha )}t^{\alpha }+\frac{a_{2}}{\Gamma (1+2\alpha )}t^{2\alpha }\right)\left(c_{0}+\frac{c_{1}}{\Gamma (1+\alpha )}t^{\alpha }+\frac{c_{2}}{\Gamma (1+2\alpha )}t^{2\alpha }\right)\\ \; & +p\left(c_{0}+\frac{c_{1}}{\Gamma (1+\alpha )}t^{\alpha }+\frac{c_{2}}{\Gamma (1+2\alpha )}t^{2\alpha }\right)\\ \; & +\lambda_{2}\left(c_{0}+\frac{c_{1}}{\Gamma (1+\alpha )}t^{\alpha }+\frac{c_{2}}{\Gamma (1+2\alpha )}t^{2\alpha }\right)\left(e_{0}+\frac{e_{1}}{\Gamma (1+\alpha )}t^{\alpha }+\frac{e_{2}}{\Gamma (1+2\alpha )}t^{2\alpha }\right)\\ \; & -\theta_{2}\left(d_{0}+\frac{d_{1}}{\Gamma (1+\alpha )}t^{\alpha }+\frac{d_{2}}{\Gamma (1+2\alpha )}t^{2\alpha }\right),\\ Res_{I_{L,2}}\left(t\right) & =D_{0}^{\alpha }\left(d_{0}+\frac{d_{1}}{\Gamma (1+\alpha )}t^{\alpha }+\frac{d_{2}}{\Gamma (1+2\alpha )}t^{2\alpha }\right)\\ \; & -\lambda_{2}\left(c_{0}+\frac{c_{1}}{\Gamma (1+\alpha )}t^{\alpha }+\frac{c_{2}}{\Gamma (1+2\alpha )}t^{2\alpha }\right)\left(e_{0}+\frac{e_{1}}{\Gamma (1+\alpha )}t^{\alpha }+\frac{e_{2}}{\Gamma (1+2\alpha )}t^{2\alpha }\right)\\ \; & +r\left(d_{0}+\frac{d_{1}}{\Gamma (1+\alpha )}t^{\alpha }+\frac{d_{2}}{\Gamma (1+2\alpha )}t^{2\alpha }\right)\;\;\mathrm{and}\\ Res_{L_{2}}\left(t\right) & =D_{0}^{\alpha }\left(e_{0}+\frac{e_{1}}{\Gamma (1+\alpha )}t^{\alpha }+\frac{e_{2}}{\Gamma (1+2\alpha )}t^{2\alpha }\right)\\ \; & -\mu \left(c_{0}+\frac{c_{1}}{\Gamma (1+\alpha )}t^{\alpha }+\frac{c_{2}}{\Gamma (1+2\alpha )}t^{2\alpha }\right)+\phi \left(e_{0}+\frac{e_{1}}{\Gamma (1+\alpha )}t^{\alpha }+\frac{e_{2}}{\Gamma (1+2\alpha )}t^{2\alpha }\right) \end{array}$$

We apply the operator $D_{0}^{\alpha }$ to $Res_{S_{2}}\left(t\right),Res_{S_{L,2}}\left(t\right),Res_{I_{2}}\left(t\right),Res_{I_{L,2}}\left(t\right)$ and $Res_{L_{2}}\left(t\right)$ and then evaluate the result at t=0 to get
$$\begin{array}{cc} D_{0}^{\alpha }Res_{S_{2}}\left(0\right) & =a_{2}+\beta \left[a_{0}c_{1}+a_{1}c_{0}\right]+\lambda_{1}\left[a_{0}e_{1}+a_{1}e_{0}\right]+da_{1}-\gamma_{1}c_{1}-\gamma_{2}d_{1}-\theta_{1}b_{1},\\ D_{0}^{\alpha }Res_{S_{L,2}}\left(0\right) & =b_{2}-\lambda_{1}\left[a_{0}e_{1}+a_{1}e_{0}\right]+hb_{1},\\ D_{0}^{\alpha }Res_{I_{2}}\left(0\right) & =c_{2}-\beta \left[a_{0}c_{1}+a_{1}c_{0}\right]+pc_{1}+\lambda_{2}\left[c_{0}e_{1}+c_{1}e_{0}\right]-\theta_{2}d_{1},\\ D_{0}^{\alpha }Res_{I_{L,2}}\left(0\right) & =d_{2}-\lambda_{2}\left[c_{0}e_{1}+c_{1}e_{0}\right]+rd_{1},\\ D_{0}^{\alpha }Res_{L_{2}}\left(0\right) & =e_{2}-\mu c_{1}+\phi e_{1}. \end{array}$$

By solving the equations
$$\begin{array}{c} D_{0}^{\alpha }Res_{S_{2}}\left(0\right)=0,\;\;D_{0}^{\alpha }Res_{S_{L,2}}\left(0\right)=0,\;\;D_{0}^{\alpha }Res_{I_{2}}\left(0\right)=0,\;\;D_{0}^{\alpha }Res_{I_{L,2}}\left(0\right)=0\;\mathrm{and}\;D_{0}^{\alpha }Res_{L_{2}}\left(0\right)=0, \end{array}$$
we have
$$\begin{array}{cc} a_{2} & =-\beta \left[a_{0}c_{1}+a_{1}c_{0}\right]-\lambda_{1}\left[a_{0}e_{1}+a_{1}e_{0}\right]-da_{1}+\gamma_{1}c_{1}+\gamma_{2}d_{1}+\theta_{1}b_{1},\\ b_{2} & =\lambda_{1}\left[a_{0}e_{1}+a_{1}e_{0}\right]-hb_{1},\\ c_{2} & =\beta \left[a_{0}c_{1}+a_{1}c_{0}\right]-pc_{1}-\lambda_{2}\left[c_{0}e_{1}+c_{1}e_{0}\right]+\theta_{2}d_{1},\\ d_{2} & =\lambda_{2}\left[c_{0}e_{1}+c_{1}e_{0}\right]-rd_{1},\\ e_{2} & =\mu c_{1}-\phi e_{1}. \end{array}$$

Similarly, the third truncated power series approximations are given by
$$\begin{array}{cc} S_{2}\left(t\right) & =a_{0}+\frac{a_{1}}{\Gamma (1+\alpha )}t^{\alpha }+\frac{a_{2}}{\Gamma (1+2\alpha )}t^{2\alpha }+\frac{a_{3}}{\Gamma (1+3\alpha )}t^{3\alpha },\\ S_{L,2}\left(t\right) & =b_{0}+\frac{b_{1}}{\Gamma (1+\alpha )}t^{\alpha }+\frac{b_{2}}{\Gamma (1+2\alpha )}t^{2\alpha }+\frac{b_{3}}{\Gamma (1+3\alpha )}t^{3\alpha },\\ I_{2}\left(t\right) & =c_{0}+\frac{c_{1}}{\Gamma (1+\alpha )}t^{\alpha }+\frac{c_{2}}{\Gamma (1+2\alpha )}t^{2\alpha }+\frac{c_{3}}{\Gamma (1+3\alpha )}t^{3\alpha },\\ I_{L,2}\left(t\right) & =d_{0}+\frac{d_{1}}{\Gamma (1+\alpha )}t^{\alpha }+\frac{d_{2}}{\Gamma (1+2\alpha )}t^{2\alpha }+\frac{d_{3}}{\Gamma (1+3\alpha )}t^{3\alpha },\;\;\mathrm{and}\\ L_{2}\left(t\right) & =d_{0}+\frac{e_{1}}{\Gamma (1+\alpha )}t^{\alpha }+\frac{e_{2}}{\Gamma (1+2\alpha )}t^{2\alpha }+\frac{e_{3}}{\Gamma (1+3\alpha )}t^{3\alpha }. \end{array}$$

So, the third residual functions are:$$\begin{array}{cc} Res_{S_{3}}\left(t\right) & =D_{0}^{\alpha }\left(a_{0}+\frac{a_{1}}{\Gamma (1+\alpha )}t^{\alpha }+\frac{a_{2}}{\Gamma (1+2\alpha )}t^{2\alpha }+\frac{a_{3}}{\Gamma (1+3\alpha )}t^{3\alpha }\right)-\Lambda \\ \; & +\beta \left(a_{0}+\frac{a_{1}}{\Gamma (1+\alpha )}t^{\alpha }+\frac{a_{2}}{\Gamma (1+2\alpha )}t^{2\alpha }+\frac{a_{3}}{\Gamma (1+3\alpha )}t^{3\alpha }\right)\\ \; & \times \left(c_{0}+\frac{c_{1}}{\Gamma (1+\alpha )}t^{\alpha }+\frac{c_{2}}{\Gamma (1+2\alpha )}t^{2\alpha }+\frac{c_{3}}{\Gamma (1+3\alpha )}t^{3\alpha }\right)\\ \; & +\lambda_{1}\left(a_{0}+\frac{a_{1}}{\Gamma (1+\alpha )}t^{\alpha }+\frac{a_{2}}{\Gamma (1+2\alpha )}t^{2\alpha }+\frac{a_{3}}{\Gamma (1+3\alpha )}t^{3\alpha }\right)\\ \; & \times \left(e_{0}+\frac{e_{1}}{\Gamma (1+\alpha )}t^{\alpha }+\frac{e_{2}}{\Gamma (1+2\alpha )}t^{2\alpha }+\frac{e_{3}}{\Gamma (1+3\alpha )}t^{3\alpha }\right)\\ \; & +d\left(a_{0}+\frac{a_{1}}{\Gamma (1+\alpha )}t^{\alpha }+\frac{a_{2}}{\Gamma (1+2\alpha )}t^{2\alpha }+\frac{a_{3}}{\Gamma (1+3\alpha )}t^{3\alpha }\right)\\ \; & -\gamma_{1}\left(c_{0}+\frac{c_{1}}{\Gamma (1+\alpha )}t^{\alpha }+\frac{c_{2}}{\Gamma (1+2\alpha )}t^{2\alpha }+\frac{c_{3}}{\Gamma (1+3\alpha )}t^{3\alpha }\right)\\ \; & -\gamma_{2}\left(d_{0}+\frac{d_{1}}{\Gamma (1+\alpha )}t^{\alpha }+\frac{d_{2}}{\Gamma (1+2\alpha )}t^{2\alpha }+\frac{d_{3}}{\Gamma (1+3\alpha )}t^{3\alpha }\right)\\ \; & -\theta_{1}\left(3+\frac{b_{1}}{\Gamma (1+\alpha )}t^{\alpha }+\frac{b_{2}}{\Gamma (1+2\alpha )}t^{2\alpha }+\frac{b_{3}}{\Gamma (1+3\alpha )}t^{3\alpha }\right),\\ Res_{S_{L,3}}\left(t\right) & =D_{0}^{\alpha }\left(b_{0}+\frac{b_{1}}{\Gamma (1+\alpha )}t^{\alpha }+\frac{b_{2}}{\Gamma (1+2\alpha )}t^{2\alpha }+\frac{b_{3}}{\Gamma (1+3\alpha )}t^{3\alpha }\right)\\ \; & -\lambda_{1}\left(a_{0}+\frac{a_{1}}{\Gamma (1+\alpha )}t^{\alpha }+\frac{a_{2}}{\Gamma (1+2\alpha )}t^{2\alpha }+\frac{a_{3}}{\Gamma (1+3\alpha )}t^{3\alpha }\right)\\ \; & \times \left(e_{0}+\frac{e_{1}}{\Gamma (1+\alpha )}t^{\alpha }+\frac{e_{2}}{\Gamma (1+2\alpha )}t^{2\alpha }+\frac{e_{3}}{\Gamma (1+3\alpha )}t^{3\alpha }\right)\\ \; & +h\left(b_{0}+\frac{b_{1}}{\Gamma (1+\alpha )}t^{\alpha }+\frac{b_{2}}{\Gamma (1+2\alpha )}t^{2\alpha }+\frac{b_{3}}{\Gamma (1+3\alpha )}t^{3\alpha }\right),\\ Res_{I_{3}}\left(t\right) & =D_{0}^{\alpha }\left(c_{0}+\frac{c_{1}}{\Gamma (1+\alpha )}t^{\alpha }+\frac{c_{2}}{\Gamma (1+2\alpha )}t^{2\alpha }+\frac{c_{3}}{\Gamma (1+3\alpha )}t^{3\alpha }\right)\\ \; & -\beta \left(a_{0}+\frac{a_{1}}{\Gamma (1+\alpha )}t^{\alpha }+\frac{a_{2}}{\Gamma (1+2\alpha )}t^{2\alpha }+\frac{a_{3}}{\Gamma (1+3\alpha )}t^{3\alpha }\right)\\ \; & \times \left(c_{0}+\frac{c_{1}}{\Gamma (1+\alpha )}t^{\alpha }+\frac{c_{2}}{\Gamma (1+2\alpha )}t^{2\alpha }+\frac{c_{3}}{\Gamma (1+3\alpha )}t^{3\alpha }\right)\\ \; & +p\left(c_{0}+\frac{c_{1}}{\Gamma (1+\alpha )}t^{\alpha }+\frac{c_{2}}{\Gamma (1+2\alpha )}t^{2\alpha }+\frac{c_{3}}{\Gamma (1+3\alpha )}t^{3\alpha }\right)\\ \; & +\lambda_{2}\left(c_{0}+\frac{c_{1}}{\Gamma (1+\alpha )}t^{\alpha }+\frac{c_{2}}{\Gamma (1+2\alpha )}t^{2\alpha }+\frac{c_{3}}{\Gamma (1+3\alpha )}t^{3\alpha }\right)\\ \; & \times \left(e_{0}+\frac{e_{1}}{\Gamma (1+\alpha )}t^{\alpha }+\frac{e_{2}}{\Gamma (1+2\alpha )}t^{2\alpha }+\frac{e_{3}}{\Gamma (1+3\alpha )}t^{3\alpha }\right)\\ \; & -\theta_{2}\left(d_{0}+\frac{d_{1}}{\Gamma (1+\alpha )}t^{\alpha }+\frac{d_{2}}{\Gamma (1+2\alpha )}t^{2\alpha }+\frac{d_{3}}{\Gamma (1+3\alpha )}t^{3\alpha }\right),\\ Res_{I_{L,3}}\left(t\right) & =D_{0}^{\alpha }\left(d_{0}+\frac{d_{1}}{\Gamma (1+\alpha )}t^{\alpha }+\frac{d_{2}}{\Gamma (1+2\alpha )}t^{2\alpha }+\frac{d_{3}}{\Gamma (1+3\alpha )}t^{3\alpha }\right)\\ \; & -\lambda_{2}\left(c_{0}+\frac{c_{1}}{\Gamma (1+\alpha )}t^{\alpha }+\frac{c_{2}}{\Gamma (1+2\alpha )}t^{2\alpha }+\frac{c_{3}}{\Gamma (1+3\alpha )}t^{3\alpha }\right)\\ \; & \times \left(e_{0}+\frac{e_{1}}{\Gamma (1+\alpha )}t^{\alpha }+\frac{e_{2}}{\Gamma (1+2\alpha )}t^{2\alpha }+\frac{e_{3}}{\Gamma (1+3\alpha )}t^{3\alpha }\right)\\ \; & +r\left(d_{0}+\frac{d_{1}}{\Gamma (1+\alpha )}t^{\alpha }+\frac{d_{2}}{\Gamma (1+2\alpha )}t^{2\alpha }+\frac{d_{3}}{\Gamma (1+3\alpha )}t^{3\alpha }\right)\;\;\mathrm{and}\\ Res_{L_{3}}\left(t\right) & =D_{0}^{\alpha }\left(e_{0}+\frac{e_{1}}{\Gamma (1+\alpha )}t^{\alpha }+\frac{e_{2}}{\Gamma (1+2\alpha )}t^{2\alpha }+\frac{e_{3}}{\Gamma (1+3\alpha )}t^{3\alpha }\right)\\ \; & -\mu \left(c_{0}+\frac{c_{1}}{\Gamma (1+\alpha )}t^{\alpha }+\frac{c_{2}}{\Gamma (1+2\alpha )}t^{2\alpha }+\frac{c_{3}}{\Gamma (1+3\alpha )}t^{3\alpha }\right)\\ \; & +\phi \left(e_{0}+\frac{e_{1}}{\Gamma (1+\alpha )}t^{\alpha }+\frac{e_{2}}{\Gamma (1+2\alpha )}t^{2\alpha }+\frac{e_{3}}{\Gamma (1+3\alpha )}t^{3\alpha }\right) \end{array}$$

We apply the operator $D_{0}^{2\alpha }$ to $Res_{S_{3}}\left(t\right),Res_{S_{L,3}}\left(t\right),Res_{I_{3}}\left(t\right),Res_{I_{L,3}}\left(t\right)$ and $Res_{L_{3}}\left(t\right)$ and then evaluate the result at t=0 to get
$$\begin{array}{cc} D_{0}^{2\alpha }Res_{S_{3}}\left(0\right) & =a_{3}+\beta \left[a_{0}c_{2}+\frac{a_{1}c_{1}\Gamma \left(1+2\alpha \right)}{\Gamma {\left(1+\alpha \right)}^{2}}+a_{2}c_{0}\right]+\lambda_{1}\left[a_{0}e_{2}+\frac{a_{1}e_{1}\Gamma \left(1+2\alpha \right)}{\Gamma {\left(1+\alpha \right)}^{2}}+a_{2}e_{0}\right]\\ \; & \;\;\;\;\;\;\;\;\;\;\;\;\;\;\;\;\;\;+da_{2}-\gamma_{1}c_{2}-\gamma_{2}d_{2}-\theta_{1}b_{2},\\ D_{0}^{2\alpha }Res_{S_{L,3}}\left(0\right) & =b_{3}-\lambda_{1}\left[a_{0}e_{2}+\frac{a_{1}e_{1}\Gamma \left(1+2\alpha \right)}{\Gamma {\left(1+\alpha \right)}^{2}}+a_{2}e_{0}\right]+hb_{2},\\ D_{0}^{2\alpha }Res_{I_{3}}\left(0\right) & =c_{3}-\beta \left[a_{0}c_{2}+\frac{a_{1}c_{1}\Gamma \left(1+2\alpha \right)}{\Gamma {\left(1+\alpha \right)}^{2}}+a_{2}c_{0}\right]+pc_{2}\\ \; & \;\;\;\;\;\;\;\;\;\;\;\;\;\;\;+\lambda_{2}\left[c_{0}e_{2}+\frac{c_{1}e_{1}\Gamma \left(1+2\alpha \right)}{\Gamma {\left(1+\alpha \right)}^{2}}+c_{2}e_{0}\right]-\theta_{2}d_{2},\\ D_{0}^{2\alpha }Res_{I_{L,3}}\left(0\right) & =d_{3}-\lambda_{2}\left[c_{0}e_{2}+\frac{c_{1}e_{1}\Gamma \left(1+2\alpha \right)}{\Gamma {\left(1+\alpha \right)}^{2}}+c_{2}e_{0}\right]+rd_{2},\\ D_{0}^{2\alpha }Res_{L_{3}}\left(0\right) & =e_{3}-\mu c_{2}+\phi e_{2}. \end{array}$$

By solving the equations
$$\begin{array}{c} D_{0}^{\alpha }Res_{S_{3}}\left(0\right)=0,\;\;D_{0}^{\alpha }Res_{S_{L,3}}\left(0\right)=0,\;\;D_{0}^{\alpha }Res_{I_{3}}\left(0\right)=0,\;\;D_{0}^{\alpha }Res_{I_{L,3}}\left(0\right)=0\;\mathrm{and}\;D_{0}^{\alpha }Res_{L_{3}}\left(0\right)=0, \end{array}$$
we have
$$\begin{array}{cc} a_{3} & =-\beta \left[a_{0}c_{2}+\frac{a_{1}c_{1}\Gamma \left(1+2\alpha \right)}{\Gamma {\left(1+\alpha \right)}^{2}}+a_{2}c_{0}\right]-\lambda_{1}\left[a_{0}e_{2}+\frac{a_{1}e_{1}\Gamma \left(1+2\alpha \right)}{\Gamma {\left(1+\alpha \right)}^{2}}+a_{2}e_{0}\right]\\ \; & \;\;\;\;\;\;\;\;\;\;\;\;\;\;\;\;\;\;-da_{2}+\gamma_{1}c_{2}+\gamma_{2}d_{2}+\theta_{1}b_{2},\\ b_{3} & =\lambda_{1}\left[a_{0}e_{2}+\frac{a_{1}e_{1}\Gamma \left(1+2\alpha \right)}{\Gamma {\left(1+\alpha \right)}^{2}}+a_{2}e_{0}\right]-hb_{2},\\ c_{3} & =\beta \left[a_{0}c_{2}+\frac{a_{1}c_{1}\Gamma \left(1+2\alpha \right)}{\Gamma {\left(1+\alpha \right)}^{2}}+a_{2}c_{0}\right]-\lambda_{2}\left[c_{0}e_{2}+\frac{c_{1}e_{1}\Gamma \left(1+2\alpha \right)}{\Gamma {\left(1+\alpha \right)}^{2}}+c_{2}e_{0}\right]\\ \; & \;\;\;\;\;\;\;\;\;\;\;\;\;\;\;-pc_{2}+\theta_{2}d_{2},\\ d_{3} & =\lambda_{2}\left[c_{0}e_{2}+\frac{c_{1}e_{1}\Gamma \left(1+2\alpha \right)}{\Gamma {\left(1+\alpha \right)}^{2}}+c_{2}e_{0}\right]-rd_{2},\\ e_{3} & =\mu c_{2}-\phi e_{2}. \end{array}$$

By continuing in this direction, we deduce that the coefficients $a_{n},b_{n},c_{n},d_{n}$ and $e_{n}$ for n≥2 are given by the recursive formula
(32)$$\begin{array}{cc} a_{n} & =-\beta \Gamma \left(1+\left(n-1\right)\alpha \right)\sum_{i=0}^{n-1}\frac{c_{n-1-i}a_{i}}{\Gamma (1+i\alpha )\Gamma (1+(n-1-i)\alpha )}\\ \; & \;\;\;-\lambda_{1}\Gamma \left(1+\left(n-1\right)\alpha \right)\sum_{i=0}^{n-1}\frac{e_{n-1-i}a_{i}}{\Gamma (1+i\alpha )\Gamma (1+(n-1-i)\alpha )}\\ \; & \;\;\;-da_{n-1}+\gamma_{1}c_{n-1}+\gamma_{2}d_{n-1}+\theta_{1}b_{n-1},\\ b_{n} & =\lambda_{1}\Gamma \left(1+\left(n-1\right)\alpha \right)\sum_{i=0}^{n-1}\frac{e_{n-1-i}a_{i}}{\Gamma (1+i\alpha )\Gamma (1+(n-1-i)\alpha )}-hb_{n-1},\\ c_{n} & =\beta \Gamma \left(1+\left(n-1\right)\alpha \right)\sum_{i=0}^{n-1}\frac{c_{n-1-i}a_{i}}{\Gamma (1+i\alpha )\Gamma (1+(n-1-i)\alpha )}\\ \; & \;\;-\lambda_{2}\Gamma \left(1+\left(n-1\right)\alpha \right)\sum_{i=0}^{n-1}\frac{e_{n-1-i}c_{i}}{\Gamma (1+i\alpha )\Gamma (1+(n-1-i)\alpha )}\\ \; & \;\;-pc_{n-1}+\theta_{2}d_{n-1},\\ d_{n} & =\lambda_{2}\Gamma \left(1+\left(n-1\right)\alpha \right)\sum_{i=0}^{n-1}\frac{e_{n-1-i}c_{i}}{\Gamma (1+i\alpha )\Gamma (1+(n-1-i)\alpha )}-rd_{n-1},\\ e_{n} & =\mu c_{n-1}-\phi e_{n-1}. \end{array}$$

**Remark** **7.**
*It is worth noting that these recursive formulas for the coefficients will ensure that we obtain higher-order approximate solutions as compared with similar results in the literature with a smaller order of the approximate solutions. Thus, due to this recursive nature of the coefficients, we can calculate $a_{n},b_{n},c_{n},d_{n}$ and $e_{n}$ for any large n values if necessary. For the purpose of the numerical simulation given in Section 6, we used n=25. That is, the coefficients $a_{n},b_{n},c_{n},d_{n},e_{n}$ for n=1,2,3,⋯25 are computed and the approximated fractional power series solutions of system (Equation 8) are plotted in Section 6, numerical simulation and examples.*


## 6. Numerical Simulations and Examples

The following table provides the description of the different parameters used in the COVID-19 model. The parameter values under column “data set 1” yield a basic reproduction number $R_{0}=0.2373$. Similarly, if the parameters under the column “data set 2” are used, then we have $R_{0}=4.5707$. In Section 6.1, we simulate both the exact and approximated solutions of system (Equation 8) using the values under “data set 1” of Table 2 for various α values (α=0.5,0.75,0.95 and 1). In Section 6.2, we simulate the exact and approximated solutions of system (Equation 8) using the parameter values under “data set 2” of table Table 2. Moreover, in both cases ($R_{0}<1\;\mathrm{and}\;R_{0}>1$) the relative errors, due to approximating the exact solution of system (Equation 8) by the residual power series method, are computed for different α values as indicated in Table 3 and Table 4.

### 6.1. Exact and Approximated Solutions of System (Equation 8) When $R_{0}<1$

Using data set 1, the basic reproduction number $R_{0}$ is calculated to be 0.2373. Thus by Theorems 4 and 5, the disease-free equilibrium point is both locally and globally asymptotically stable. The disease-free equilibrium point is calculated to be $E_{0}=\left(4167,0,0,0,0\right)$. The simulations in Figure 1 and Figure 2 support the fact that the trajectories of system (Equation 8) converges to this equilibrium point $E_{0}=\left(4167,0,0,0,0\right)$.

In Figure 1 and Figure 2 the exact and approximated solutions of system (Equation 8) are plotted for 0≤t≤100, respectively. In order to show the accuracy of the residual power series method for approximating the solution of system (Equation 8), we have calculated the relative error of each state variable for various α values. A time step size $\Delta t=2^{-6}$ is considered and for each state variables $x\left(t\right)\in \{S\left(t\right),S_{L}\left(t\right),I\left(t\right),I_{L}\left(t\right),L\left(t\right)\}$ the values x(iΔt) are calculated where 0≤i≤6400. The relative error for the state variable x(t) is computed using
(33)$$Rel_{x(t)}=\max_{0\leq i\leq 6400}\left\{\frac{\left|Exact(x(i\Delta t))-Appr(x(i\Delta t))\right|}{Exact(x(i\Delta t))}\right\},$$
where Exact(x(iΔt)) and Appr(x(iΔt)) represent the exact and approximated values of the variable x(t) at t=iΔt for 0≤i≤6400. As can be seen from Table 3, the relative error of each state variable decreases as the value of α increases from 0.5 to 1.

### 6.2. Exact and Approximated Solutions of System (Equation 8) When $R_{0}>1$

The basic reproduction number is calculated using the parameter values under data set 2. The value of $R_{0}$ is found to be 4.5707, and also the condition $\lambda_{2}<\theta_{2}$ is satisfied. Thus by Remark 6 and Theorem 7 the endemic equilibrium point $E_{1}=\left(1458,0,3370,0,0\right)$, which is calculated using Equation (Equation 10), is both locally and globally asymptotically stable. Note that both Figure 3 and Figure 4 show that the trajectories of the solution of system (Equation 8) converges to the equilibrium point $E_{1}$.

Similar to the previous discussion, the relative errors of the state variables are calculated for various α values using Equation (Equation 33).

## 7. Conclusions and Discussion

In this paper, a fractional-order COVID-19 epidemic model with lockdown is proposed and analyzed. In particular, some of the most important epidemiological constants, such as the equilibrium points ($E_{0},E_{1}$ and $E_{2}$) and the basic reproduction number $R_{0}$, are calculated. The existence and uniqueness of a positive global solution of the fractional-order system (Equation 8) is established. Moreover, by implementing various techniques such as the comparison theory of fractional-order differential equations, Mittage-Leffler stability (Theorem 5), and fractional La-Salle invariance principle (Theorem 6), the local and global stabilities of the equilibrium points are discussed. Additionally, the method of residual power series is used to approximate the solution of system (Equation 8). As can be seen in Equation (Equation 32), the coefficients $a_{n},b_{n},c_{n},d_{n}$ and $e_{n}$ have recursive nature. That is, for any n≥1, each coefficients $a_{n},b_{n},c_{n},d_{n}$ and $e_{n}$ can be determined from the previous ones $a_{n-1},b_{n-1},c_{n-1},d_{n-1}$ and $e_{n-1}$. Thus we can calculate as many coefficients as we need in order to get a very good approximation of the solution of system (Equation 8). Moreover, it is very interesting to note that as the value of α increases from 0.5 to 1, the trajectories of the solution of the fractional-order system are converging to the case where α=1. That is, the solutions of the fractional-order differential equation converge to the solutions of the system of the ordinary differential equation as the order α increases towards 1. In Section 5, numerical simulations and examples are presented to show the stability of the equilibrium points when $R_{0}<1$ and $R_{0}>1.$ The exact solutions of system (Equation 8) for various α values are shown in Figure 1 and Figure 3. Similarly, the approximated solutions of system (Equation 8) using the residual power series method are included in Figure 2 and Figure 4. Finally, in order to see the accuracy of the approximated solutions, we provided the relative errors for different α values as shown in Table 3 and Table 4. The model that is studied in this paper can be used to predict and understand how COVID-19 infection spreads in the presence of a lockdown. In addition to understanding the dynamics of the infection, it is very important to determine the relative importance of the various parameters used in the model. One such technique is the study of the sensitivity of the parameters in relation to the basic reproduction number $R_{0}$, and the two endemic equilibrium points $E_{1}$ and $E_{2}$, which are considered to be the most important values in the study of deterministic epidemic models. One such approach is to calculate the normalized sensitivity index (NSI), denoted by $SI_{\kappa }$, where κ is the given parameter. For example, if κ is one of the parameters in $R_{0}$, then the NSI of κ is defined as $SI_{\kappa }=\frac{\kappa }{R_{0}}\frac{\partial R_{0}}{\partial \kappa }.$ Using this result, we have calculated the NSI of the parameters β and $\gamma_{1},$ which are directly affected by the infection. As a result, we have $SI_{\beta }=1$, and $SI_{\gamma_{1}}=\frac{-\gamma_{1}}{\gamma_{1}+\alpha_{1}+d}<0.$ Thus in order to decrease $R_{0}$, the expected number of infections generated by one case, we have to either decrease the infection contact rate β or increase the recovery rate of the infected group $\gamma_{1}.$ This further implies imposing some prevention actions, such as avoiding contact with people who are infected, wearing masks, and keeping a distance from an infectious person, will help to decrease the value of $R_{0}$, which directly leads to mitigating the infection. By modifying this model, one can study other aspects of the infection, such as the impact of vaccination, the effect of population diffusion, and other important factors that determine the transmission and persistence of the infection.

## Figures and Tables

**Figure 1 vaccines-10-01773-f001:**
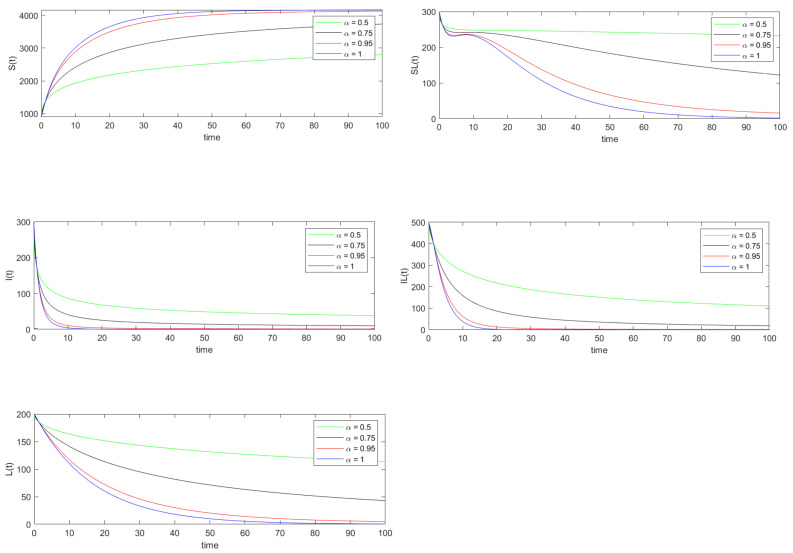
Numerical simulation of the trajectories of the solution of system (Equation 8) when $R_{0}<1.$

**Figure 2 vaccines-10-01773-f002:**
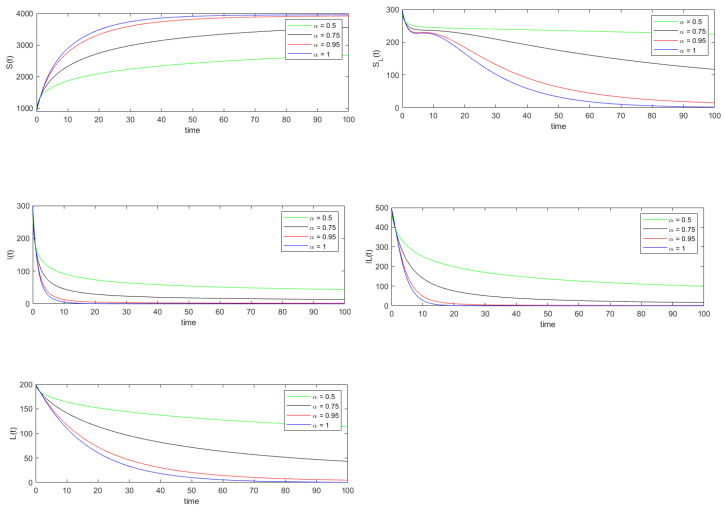
Approximated solution of system (Equation 8) using the residual power series method when $R_{0}<1.$ Note that the values of the parameters used in this figure are the same as the one in Figure 1.

**Figure 3 vaccines-10-01773-f003:**
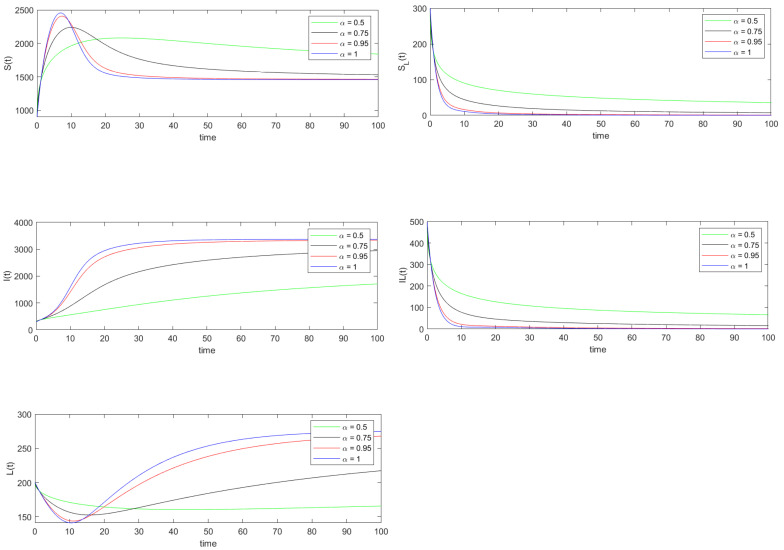
Numerical simulation of the trajectories of the solution of system (Equation 8) when $R_{0}>1.$

**Figure 4 vaccines-10-01773-f004:**
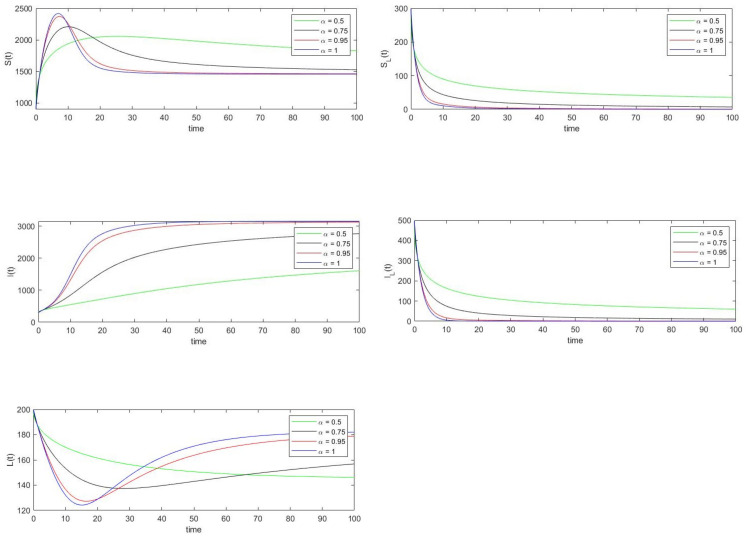
Approximated solution of system (Equation 8) using the residual power series method when $R_{0}>1.$ The values of the parameters used in this figure are the same as in Figure 3.

**Table 1 vaccines-10-01773-t001:** Description of the parameters used in the COVID-19 epidemic model (Equation 8).

Parameters	Description of the Parameters
Λ	Recruitment rate
β	Infection contact rate
$\lambda_{1}$	Imposition of lockdown on susceptible group
$\lambda_{2}$	Imposition of lockdown on infected group
$\gamma_{1}$	Recovery rate of the infected group
$\gamma_{2}$	Recovery rate of the infected group under lockdown
$\alpha_{1}$	Death rate of the infected group
$\alpha_{2}$	Death rate of the infected group under lockdown
*d*	Natural death rates
$\theta_{1}$	Rate of transfer of susceptible lockdown individuals to susceptible class
$\theta_{2}$	Rate of transfer of susceptible lockdown individuals to infected class
μ	rate of implementation of the lockdown program
ϕ	rate of depletion of the lockdown program

**Table 2 vaccines-10-01773-t002:** Input parameter values used to simulate the trajectories of the solution of the model as shown in Figure 1, Figure 2, Figure 3 and Figure 4. The parameters under the column data set 1 will result $R_{0}=0.2373<1$, and the parameters under data set 2 yield $R_{0}=4.5707>1.$

Parameters	Description of the Parameters	Data Set 1	Data Set 2	Reference
Λ	Recruitment rate	400	400	[38,43]
β	Infection contact rate	$1.7\times 10^{-5}$	$1.8\times 10^{-4}$	[44]
$\lambda_{1}$	Imposition of lockdown on susceptible group	$2\times 10^{-4}$	$2\times 10^{-5}$	[38,43]
$\lambda_{2}$	Imposition of lockdown on infected group	0.002	0.002	[38,43]
$\gamma_{1}$	Recovery rate of the infected group	0.16979	0.16979	[38,43]
$\gamma_{2}$	Recovery rate of the infected group under lockdown	0.16979	0.16979	[38,43]
$\alpha_{1}$	Death rate of the infected group	0.03275	0.03275	[44]
$\alpha_{2}$	Death rate of the infected group under lockdown	0.03275	0.03275	[44]
*d*	Natural death rates	0.096	0.06	[38,43]
$\theta_{1}$	Rate of transfer of susceptible lockdown individuals to susceptible class	0.2	0.52	[38,43]
$\theta_{2}$	Rate of transfer of susceptible lockdown individuals to infected class	0.02	0.2	[38,43]
μ	rate of implementation of the lockdown program	$5\times 10^{-4}$	$5\times 10^{-5}$	[38,43]
ϕ	rate of depletion of the lockdown program	0.06	0.06	[38,43]

**Table 3 vaccines-10-01773-t003:** Relative error of the state variables for different values of α when $R_{0}<1$.

	α=0.5	α=0.75	α=0.95	α=1
*S*	$3.876\times 10^{-6}$	$3.143\times 10^{-6}$	$2.025\times 10^{-6}$	$1.202\times 10^{-6}$
$S_{L}$	$4.302\times 10^{-6}$	$2.710\times 10^{-6}$	$7.094\times 10^{-7}$	$4.851\times 10^{-7}$
*I*	$6.813\times 10^{-7}$	$2.421\times 10^{-7}$	$8.059\times 10^{-8}$	$3.140\times 10^{-8}$
$I_{L}$	$1.783\times 10^{-6}$	$0.448\times 10^{-6}$	$2.220\times 10^{-7}$	$1.504\times 10^{-7}$
*L*	$1.570\times 10^{-6}$	$2.097\times 10^{-6}$	$1.313\times 10^{-6}$	$9.177\times 10^{-7}$

**Table 4 vaccines-10-01773-t004:** Relative error of the state variables for different values of α when $R_{0}>1$.

	α=0.5	α=0.75	α=0.95	α=1
*S*	$4.013\times 10^{-6}$	$1.571\times 10^{-6}$	$3.014\times 10^{-7}$	$2.624\times 10^{-7}$
$S_{L}$	$8.201\times 10^{-5}$	$2.930\times 10^{-5}$	$5.466\times 10^{-6}$	$2.181\times 10^{-6}$
*I*	$1.099\times 10^{-6}$	$8.205\times 10^{-7}$	$3.272\times 10^{-7}$	$6.142\times 10^{-8}$
$I_{L}$	$4.240\times 10^{-6}$	$3.279\times 10^{-6}$	$1.142\times 10^{-6}$	$8.529\times 10^{-7}$
*L*	$5.014\times 10^{-5}$	$1.792\times 10^{-5}$	$7.263\times 10^{-6}$	$1.817\times 10^{-6}$

## Data Availability

Not applicable.

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
