# Peer review of "Analysis of a Fractional-Order COVID-19 Epidemic Model with Lockdown"

_vaccines, 2022, doi:10.3390/vaccines10111773_

Round 1
Reviewer 1 Report
Fractional calculus is a powerful tool that has been used in several contexts. In fact, with a simple extension of the differential operators, it is possible to obtain various behaviors and model different aspects of the systems. The present paper uses fractional calculus to extend an epidemic model and analyses it for Covid-19. The authors studied the local and global stability of the disease-free equilibrium and endemic equilibrium solutions. In addition, they obtained a fractional power series approximation of the analytic solution. The results are interesting and relevant to the field. My suggestion for the authors is to improve the conclusions with additional discussions about their model in comparison with other models. Another minor point is about the style of the manuscript, which can be improved, for example, removing the words “affiliations”, the addresses are enough. Some references have an extra "[...]" and others.
Author Response
Dear reviewer,
We would like to thank you for taking the time and effort necessary to review the
manuscript. We sincerely appreciate all valuable comments and suggestions,
which helped us to improve the quality of the manuscript. We have stated the
reviewers comment and provided a response to each of the comments. We have
tried our best to indicate what we did and mention the page number and location
of the change. We rearranged some of the text with slight modifications to
address some of the reviewers concerns and the changes that we made in the
paper are colored blue. If you need any information or have any question, please
let us know.
1. My suggestion for the authors is to improve the conclusions with additional
discussions about their model in comparison with other models.
Response: We have added some explanations in the discussion/conclusion
part as well as the introduction part to emphasize the importance of the
suggested model. Indeed, there are several other models on COVID‐19 in the
literature but this is the only model with lockdown in the literature to the best
of our knowledge.
2. Another minor point is about the style of the manuscript, which can be
improved, for example, removing the words “affiliations”, the addresses
are enough.
Response: We have edited this part as suggested.
3. Some references have an extra "[...]" and others.
Response: We have edited how the references are written and corrected the
equation numbers in the paper.
Reviewer 2 Report
This paper studies a mathematical model of the coronavirus disease with lockdown by employing the Caputo fractional-order derivative.
I suggest the publication of it after addressing the following issues:
1- There are multible fractional derivative operators. What are the reasons to use the Caputo fractional derivative? I suggest the Refs. (https://doi.org/10.1186/s13662-017-1306-z, https://doi.org/10.1186/s13662-021-03547-x) to provide an overview
2- The equation number in the introduction section is missing.
3- For the Refs. citation, [[24]] should be [24], please fix this problem in the whole paper.
4- The model formulation (Section 1) needs more attention. Describe the parameters.
5- The References Section should be improved and written in the MDPI style.
Author Response
Dear reviewer,
We would like to thank you for taking the time and effort to review the
manuscript. We sincerely appreciate all valuable comments and suggestions,
which helped us to improve the quality of the manuscript. We have stated the
reviewers comment and provided a response to each of the comments. We have
tried our best to indicate what we did and mention the page number and location
of the change. We rearranged some of the text with slight modifications to
address some of the reviewers concerns and the changes that we made in the
paper are colored blue. If you need any information or have any question, please
let us know.
1‐ There are multiple fractional derivative operators. What are the reasons to use
the Caputo fractional derivative? I suggest the Refs.
(https://doi.org/10.1186/s13662‐017‐1306‐z, https://doi.org/10.1186/s13662‐
021‐03547‐x) to provide an overview
Response: We use the Caputo derivative because it possesses the so‐called
memory effect and thus it ensures that the past history is included in the model.
Indeed, the Caputo derivative is not the only fractional derivative with this
property but the fact that the Caputo derivative of a constant function is zero
among other properties makes it easier to work with especially when finding the
approximate solutions. We believe that these may also be the reasons why the
Caputo derivative is the most used in modeling. We have made some minor
revisions in the manuscript to give some clarity to this question. Please look at
page 1 and page 3.
2‐ The equation number in the introduction section is missing.
Response: We have added equation number to the equations in the introduction
part.
3‐ For the Refs. citation, [[24]] should be [24], please fix this problem in the whole
paper.
Response: We have fixed all the references as suggested.
4‐ The model formulation (Section 1) needs more attention. Describe the
parameters.
Response: We have added an explanation of the parameters used in the model.
Please look at table 1 on page 4.
5‐ The References Section should be improved and written in the MDPI style.
Response: We have rewritten the reference section following the MDPI style.